# From Tempered to Benign Overfitting
# in ReLU Neural Networks

**Guy Kornowski**
Weizmann Institute of Science
`guy.kornowski@weizmann.ac.il`

**Gilad Yehudai**
Weizmann Institute of Science
`gilad.yehudai@weizmann.ac.il`

**Ohad Shamir**
Weizmann Institute of Science
`ohad.shamir@weizmann.ac.il`

## Abstract

Overparameterized neural networks (NNs) are observed to generalize well even when trained to perfectly fit noisy data. This phenomenon motivated a large body of work on "benign overfitting", where interpolating predictors achieve near-optimal performance. Recently, it was conjectured and empirically observed that the behavior of NNs is often better described as "tempered overfitting", where the performance is non-optimal yet also non-trivial, and degrades as a function of the noise level. However, a theoretical justification of this claim for non-linear NNs has been lacking so far. In this work, we provide several results that aim at bridging these complementing views. We study a simple classification setting with 2-layer ReLU NNs, and prove that under various assumptions, the type of overfitting transitions from tempered in the extreme case of one-dimensional data, to benign in high dimensions. Thus, we show that the input dimension has a crucial role on the overfitting profile in this setting, which we also validate empirically for intermediate dimensions. Overall, our results shed light on the intricate connections between the dimension, sample size, architecture and training algorithm on the one hand, and the type of resulting overfitting on the other hand.

## 1   Introduction

Overparameterized neural networks (NNs) are observed to generalize well even when trained to perfectly fit noisy data. Although quite standard in deep learning, the so-called interpolation learning regime challenges classical statistical wisdom regarding overfitting, and has attracted much attention in recent years.

In particular, the phenomenon commonly referred to as "benign overfitting" [Bartlett et al., 2020] describes a situation in which a learning method overfits, in the sense that it achieves zero training error over inherently noisy samples, yet it is able to achieve near-optimal accuracy with respect to the underlying distribution. So far, much of the theoretical work studying this phenomenon focused on linear (or kernel) regression problems using the squared loss, with some works extending this to classification problems. However, there is naturally much interest in gradually pushing this research towards neural networks. For example, Frei et al. [2023] recently showed that the implicit bias of gradient-based training algorithms towards margin maximization, as established in a series of works throughout the past few years, leads to benign overfitting of leaky ReLU networks in high dimensions (see discussion of related work below).

Recently, Mallinar et al. [2022] suggested a more nuanced view, coining the notion of "tempered" overfitting. An interpolating learning method is said to overfit in a tempered manner if its error (with

37th Conference on Neural Information Processing Systems (NeurIPS 2023).

respect to the underlying distribution, and as the training sample size increases) is bounded away from the optimal possible error yet is also not "catastrophic", e.g., better than a random guess. To be more concrete, consider a binary classification setting (with labels in $\{\pm 1\}$), and a data distribution $\mathcal{D}$, where the output labels correspond to some ground truth function $f^*$ belonging to the predictor class of interest, corrupted with independent label noise at level $p \in (0, \frac{1}{2})$ (i.e., given a sampled point $\mathbf{x}$, its associated label $y$ equals $\text{sign}(f^*(\mathbf{x}))$ with probability $1 - p$, and $-\text{sign}(f^*(\mathbf{x}))$ with probability $p$). Suppose furthermore that the training method is such that the learned predictor achieves zero error on the (inherently noisy) training data. Finally, let $L_{\text{cl}}$ denote the "clean" test error, namely $L_{\text{cl}}(N) = \text{Pr}_{\mathbf{x} \sim \mathcal{D}}(N(\mathbf{x}) \cdot f^*(\mathbf{x}) \leq 0)$ for some predictor $N$ (omitting the contribution of the label noise). In this setting, benign, catastrophic or tempered overfitting corresponds to a situation where the clean test error $L_{\text{cl}}$ of the learned predictor approaches $0$, $\frac{1}{2}$, or some value in $(0, \frac{1}{2})$ respectively. In the latter case, the clean test error typically scales with the amount of noise $p$. For example, for the one-nearest-neighbor algorithm, it is well-known that the test error asymptotically converges to $2p(1 - p)$ in our setting [Cover and Hart, 1967], which translates to a clean test error of $p$. Mallinar et al. [2022] show how a similar behavior (with the error scaling linearly with the noise level) occurs for kernel regression, and provided empirical evidence that the same holds for neural networks.

The starting point of our paper is an intriguing experiment from that paper (Figure 6b), where they considered the following extremely simple data distributioin: The inputs are drawn uniformly at random from the unit sphere, and $f^*$ is the constant $+1$ function. This is perhaps the simplest possible setting where benign or tempered overfitting in binary classification may be studied. In this setting, the authors show empirically that a three-layer vanilla neural network exhibits tempered overfitting, with the clean test error almost linear in the label noise level $p$. Notably, the experiment used an input dimension of 10, which is rather small. This naturally leads to the question of whether we can rigorously understand the overfitting behavior of neural networks in this simple setup, and how problem parameters such as the input dimension, number of samples and the architecture affect the type of overfitting.

**Our contributions:** In this work we take a step towards rigorously understating the regimes under which simple, 2-layer ReLU neural networks exhibit different types of overfitting (i.e. benign, catastrophic or tempered), depending on the problem parameters, for the simple and natural data distribution described in the previous paragraph. We focus on interpolating neural networks trained with exponentially-tailed losses, which are well-known to converge to KKT points of a max-margin problem (see Section 2 for more details). At a high level, our main conclusion is that for such networks, the input dimension plays a crucial role, with the type of overfitting gradually transforming from tempered to benign as the dimension increases. In contrast, most of our results are not sensitive to the network's width, as long as it is sufficient to interpolate the training data. In a bit more detail, our contributions can be summarized as follows:

- **Tempered overfitting in one dimension (Theorem 3.1 and Theorem 3.2).** We prove that in one dimension (when the inputs are uniform over the unit interval), the resulting overfitting is provably tempered, with a clean test error scaling as $\Theta(\text{poly}(p))$. Moreover, under the stronger assumption that the algorithm converges to a local minimum of the max-margin problem, we show that the clean test error provably scales linearly with $p$. As far as we know, this is the first result provably establishing tempered overfitting for non-linear neural networks.

- **Benign overfitting in high dimensions (Theorem 4.1 and Theorem 4.3).** We prove that when the inputs are sampled uniformly from the unit sphere (and with the dimension scaling polynomially with the sample size), the resulting overfitting will generally be benign. In particular, we show that convergence to a max-margin predictor (up to some constant factor) implies that the clean test error decays exponentially fast to 0 with respect to the dimension. Furthermore, under an assumption on the scaling of the noise level and the network width, we obtain the same result for any KKT point of the max-margin problem.

- **The role of bias terms and catastrophic overfitting (Section 4.2).** The proof of the results mentioned earlier crucially relied on the existence of bias parameters in the network architecture. Thus, we further study the case of a bias-less network, and prove that without such biases, neural networks may exhibit catastrophic overfitting, and that in general they do not overfit benignly without further assumptions. We consider this an interesting illustration of how catastrophic overfitting is possible in neural networks, even in our simple setup.

- **Empirical study for intermediate dimensions (Section 5).** Following our theoretical results, we attempt at empirically bridging the gap between the one-dimensional and high dimensional settings. In particular, it appears that the tempered and overfitting behavior extends to a wider regime than what our theoretical results formally cover, and that the overfitting profile gradually shifts from tempered to benign as the dimension increases. This substantially extends the prior empirical observation due to Mallinar et al. [2022, Figure 6b], which exhibited tempered overfitting in input dimension 10.

We note that in an independent and concurrent work, Joshi et al. [2023] studied ReLU neural networks with one-dimensional inputs, similar to the setting we study in Section 3, and proved that tempered and catastrophic overfitting can occur for possibly non-constant target functions, depending on the loss function. However, they focus on a *regression* setting, and on min-norm predictors, whereas our results cover classification and any KKT point of the max-magin problem. Moreover, the architecture considered differs somewhat from our work, with Joshi et al. including an untrained skip connection and an untrained bias term.

## 1.1 Related work

Following the empirical observation that modern deep learning methods can perfectly fit the training data while still performing well on test data [Zhang et al., 2017], many works have tried to provide theoretical explanations of this phenomenon. By now the literature on this topic is quite large, and we will only address here the works most relevant to this paper (for a broader overview, see for example the surveys Belkin 2021, Bartlett et al. 2021, Vardi 2022).

There are many works studying regression settings in which interpolating methods, especially linear and kernel methods, succeed at obtaining optimal or near-optimal performance [Belkin et al., 2018a,b, 2019, 2020, Mei and Montanari, 2022, Hastie et al., 2022]. In particular, it is interesting to compare our results to those of Rakhlin and Zhai [2019], Beaglehole et al. [2023], who proved that minimal norm interpolating (i.e. "ridgeless") kernel methods are not consistent in any fixed dimension, while they can be whenever the dimension scales with the sample size (under suitable assumptions) [Liang and Rakhlin, 2020]. Our results highlight that the same phenomenon occurs for ReLU networks.

There is by now also a large body of work on settings under which benign overfitting (and analogous definitions thereof) occurs [Nagarajan and Kolter, 2019, Bartlett et al., 2020, Negrea et al., 2020, Nakkiran and Bansal, 2020, Yang et al., 2021, Koehler et al., 2021, Bartlett and Long, 2021, Muthukumar et al., 2021, Bachmann et al., 2021, Zhou et al., 2023, Shamir, 2023]. In classification settings, as argued by Muthukumar et al. [2021], many existing works on benign overfitting analyze settings in which max-margin predictors can be computed (or approximated), and suffice to ensure benign behavior [Poggio and Liao, 2019, Montanari et al., 2019, Thrampoulidis et al., 2020, Wang and Thrampoulidis, 2021, Wang et al., 2021, Cao et al., 2021, Hu et al., 2022, McRae et al., 2022, Liang and Recht, 2023]. It is insightful to compare this to our main proof strategy, in which we analyze the max-margin problem in the case of NNs (which no longer admits a closed form). Frei et al. [2022] showed that NNs with smoothed leaky ReLU activations overfit benignly when the data comes from a well-seperated mixture distribution, a result which was recently generalized to ReLU activations [Xu and Gu, 2023]. Similarly, Cao et al. [2022] proved that convolutional NNs with smoothed activations overfit benignly when the data is distributed according to a high dimensional Guassian with noisy labels, which was subsequently generalized to ReLU CNNs [Kou et al., 2023]. We note that these distributions are reminiscent of the setting we study in Section 4 for (non convolutional) ReLU NNs. Chatterji and Long [2023] studied benign overfitting for deep linear networks.

As mentioned in the introduction, Mallinar et al. [2022] formally introduced the notion of tempered overfitting and suggested the study of it in the context of NNs. Subsequently, Manoj and Srebro [2023] proved that the minimum description length learning rule exhibits tempered overfitting, though noticeably this learning rule does not explicitly relate to NNs. As previously mentioned, we are not aware of existing works that prove tempered overfitting in the context of NNs.

In a parallel line of work, the implicit bias of gradient-based training algorithms has received much attention [Soudry et al., 2018, Lyu and Li, 2020, Ji and Telgarsky, 2020]. Roughly speaking, these results drew the connection between NN training to margin maximization – see Section 2 for a formal reminder. For our first result (Theorem 3.1) we build upon the analysis of Safran et al. [2022] who studied this implicit bias for univariate inputs, though for the sake of bounding the number of linear

regions. In our context, Frei et al. [2023] utilized this bias to prove benign overfitting for leaky ReLU NNs in a high dimensional regime.

## 2 Preliminaries

**Notation.** We use bold-faced font to denote vectors, e.g. $\mathbf{x} \in \mathbb{R}^d$, and denote by $\|\mathbf{x}\|$ the Euclidean norm. We use $c, c', \widetilde{c}, C > 0$ etc. to denote absolute constants whose exact value can change throughout the proofs. We denote by $[n] := \{1, \ldots, n\}$, by $\mathbb{1}\{\cdot\}$ the indicator function, and by $\mathbb{S}^{d-1} \subset \mathbb{R}^d$ the unit sphere. Given sets $A \subset B$, we denote by $\mathrm{Unif}(A)$ the uniform measure over a set $A$, by $A^c$ the complementary set, and given a function $f : B \to \mathbb{R}$ we denote its restriction $f|_A : A \to \mathbb{R}$. We let $I$ be the identity matrix whenever the dimension is clear from context, and for a matrix $A$ we denote by $\|A\|$ its spectral norm. We use stadard big-O notation, with $O(\cdot), \Omega(\cdot)$ and $\Theta(\cdot)$ hiding absolute constants, write $f \lesssim g$ if $f = O(g)$, and denote by $\mathrm{poly}(\cdot)$ polynomial factors.

**Setting.** We consider a classification task based on *noisy* training data $S = (\mathbf{x}_i, y_i)_{i=1}^m \subset \mathbb{R}^d \times \{\pm 1\}$ drawn i.i.d. from an underlying distribution $\mathcal{D} := \mathcal{D}_\mathbf{x} \times \mathcal{D}_y$. Throughout, we assume that the output values $y$ are constant $+1$, corrupted with independent label noise at level $p$ (namely, each $y_i$ is independent of $\mathbf{x}_i$ and satisfies $\Pr[y_i = 1] = 1 - p, \ \Pr[y_i = -1] = p$). Note that the Bayes-optimal predictor in this setting is $f^* \equiv 1$. Given a dataset $S$, we denote by $I_+ := \{i \in [m] : y_i = 1\}$ and $I_- = \{i \in [m] : y_i = -1\}$. We study 2-layer ReLU networks:

$$N_{\boldsymbol{\theta}}(\mathbf{x}) = \sum_{j=1}^n v_j \sigma(\mathbf{w}_j \cdot \mathbf{x} + b_j),$$

where $n$ is the number of neurons or network width, $\sigma(z) := \max\{0, z\}$ is the ReLU function, $v_j, b_j \in \mathbb{R}$, $\mathbf{w}_j \in \mathbb{R}^d$ and $\boldsymbol{\theta} = (v_j, \mathbf{w}_j, b_j)_{j=1}^n \in \mathbb{R}^{(d+2)n}$ is a vectorized form of all the parameters. Throughout this work we assume that the trained network classifies the entire dataset correctly, namely $y_i N_{\boldsymbol{\theta}}(\mathbf{x}_i) > 0$ for all $i \in [m]$. Following Mallinar et al. [2022, Section 2.3] we consider the *clean* test error

$$L_{\mathrm{cl}}(N_{\boldsymbol{\theta}}) := \Pr_{\mathbf{x} \sim \mathcal{D}_\mathbf{x}} [N_{\boldsymbol{\theta}}(\mathbf{x}) \leq 0],$$

corresponding to the misclassification error with respect to the clean underlying distribution. It is said that $N_{\boldsymbol{\theta}}$ exhibits benign overfitting if $L_{\mathrm{cl}} \to 0$ with respect to large enough problem parameters (e.g. $m, d \to \infty$), while the overfitting is said to be *catastrophic* if $L_{\mathrm{cl}} \to \frac{1}{2}$. Formally, Mallinar et al. [2022] have coined tempered overfitting to describe any case in which $L_{\mathrm{cl}}$ converges to a value in $(0, \frac{1}{2})$, though a special attention has been given in the literature to cases where $L_{\mathrm{cl}}$ scales monotonically or even linearly with the noise level $p$ [Belkin et al., 2018b, Chatterji and Long, 2021, Manoj and Srebro, 2023].

**Implicit bias.** We will now briefly describe the results of Lyu and Li [2020], Ji and Telgarsky [2020] which shows that under our setting, training the network with respect to the logistic or exponential loss converges towards a KKT point of the *margin-maximization problem*. To that end, suppose $\ell(z) = \log(1 + e^{-z})$ or $\ell(z) = e^{-z}$, and consider the empirical loss $\hat{\mathcal{L}}(\boldsymbol{\theta}) = \sum_{i=1}^n \ell(y_i N_{\boldsymbol{\theta}}(\mathbf{x}_i))$. Suppose that $\boldsymbol{\theta}(t)$ evolves according to the gradient flow of the empirical loss, namely $\frac{d\boldsymbol{\theta}(t)}{dt} = -\nabla \hat{\mathcal{L}}(\boldsymbol{\theta}(t))$.[1] Note that this corresponds to performing gradient descent over the data with an infinitesimally small step size.

**Theorem 2.1** (Rephrased from Lyu and Li, 2020, Ji and Telgarsky, 2020)**.** *Under the setting above, if there exists some $t_0$ such that $\boldsymbol{\theta}(t_0)$ satisfies $\min_{i \in [m]} y_i N_{\boldsymbol{\theta}(t_0)}(\mathbf{x}_i) > 0$, then $\frac{\boldsymbol{\theta}(t)}{\|\boldsymbol{\theta}(t)\|} \overset{t \to \infty}{\longrightarrow} \frac{\boldsymbol{\theta}}{\|\boldsymbol{\theta}\|}$ for $\boldsymbol{\theta}$ which is a KKT point of the margin maximization problem*

$$\min \quad \|\boldsymbol{\theta}\|^2 \quad \text{s.t.} \quad y_i N_{\boldsymbol{\theta}}(\mathbf{x}_i) \geq 1 \quad \forall i \in [m] \ . \tag{1}$$

---

[1]Note that the ReLU function is not differentiable at 0. This can be addressed either by setting $\sigma'(0) \in [0, 1]$ as done in practical implementations, or by considering the Clarke subdifferential [Clarke, 1990] (see Lyu and Li, 2020, Dutta et al., 2013 for a discussion on non-smooth optimization). We note that this issue has no effect on our results.

The result above shows that although there are many possible parameters $\boldsymbol{\theta}$ that result in a network correctly classifying the training data, the training method has an *implicit bias* in the sense that it yields a network whose parameters are a KKT point of Problem (1). Recall that $\boldsymbol{\theta}$ is called a KKT point of Problem (1) if there exist $\lambda_1, \ldots, \lambda_m \in \mathbb{R}$ such that the following conditions hold:

$$\boldsymbol{\theta} = \sum_{i=1}^{m} \lambda_i y_i \nabla_{\boldsymbol{\theta}} N_{\boldsymbol{\theta}}(\mathbf{x}_i) \qquad \text{(stationarity)} \qquad (2)$$

$$\forall i \in [m]: \quad y_i N_{\boldsymbol{\theta}}(\mathbf{x}_i) \geq 1 \qquad \text{(primal feasibility)} \qquad (3)$$

$$\lambda_1, \ldots, \lambda_m \geq 0 \qquad \text{(dual feasibility)} \qquad (4)$$

$$\forall i \in [m]: \quad \lambda_i(y_i N_{\boldsymbol{\theta}}(\mathbf{x}_i) - 1) = 0 \qquad \text{(complementary slackness)} \qquad (5)$$

It is well-known that global or local optima of Problem (1) are KKT points, but in general, the reverse direction may not be true, even in our context of 2-layer ReLU networks (see Vardi et al., 2022). In this work, our results rely either on convergence to a KKT point (Theorem 3.1 and Theorem 4.3), or on stronger assumptions such as convergence to a local optimum (Theorem 3.2) or near-global optimum (Theorem 4.1) of Problem (1) in order to obtain stronger results.

## 3 Tempered overfitting in one dimension

Throughout this section we study the one dimensional case $d = 1$ under the uniform distribution $\mathcal{D}_x = \text{Unif}([0,1])$, so that $L_{\text{cl}}(N_{\boldsymbol{\theta}}) = \text{Pr}_{x \sim \text{Unif}([0,1])}[N_{\boldsymbol{\theta}}(x) \leq 0]$. We note that although we focus here on the uniform distribution for simplicity, our proof techniques are applicable in principle to other distributions with bounded density. Further note that both results to follow are independent of the number of neurons, so that in our setting, tempered overfitting occurs as soon as the network is wide enough to interpolate the data. We first show that any KKT point of the margin maximization problem (and thus any network we may converge to) gives rise to tempered overfitting.

**Theorem 3.1.** *Let $d = 1$, $p \in [0, \frac{1}{2})$. Then with probability at least $1 - \delta$ over the sample $S \sim \mathcal{D}^m$, for any KKT point $\boldsymbol{\theta} = (v_j, w_j, b_j)_{j=1}^n$ of Problem (1), it holds that*

$$c\left(p^5 - \sqrt{\frac{\log(m/\delta)}{m}}\right) \leq L_{\text{cl}}(N_{\boldsymbol{\theta}}) \leq C\left(\sqrt{p} + \sqrt{\frac{\log(m/\delta)}{m}}\right) ,$$

*where $c, C > 0$ are absolute constants.*

The theorem above shows that for a large enough sample size $L_{\text{cl}}(N_{\boldsymbol{\theta}}) = \Theta(\text{poly}(p))$, proving that the clean test error must scale roughly monotonically with the noise level $p$, leading to tempered overfitting.[2]

We will now provide intuition for the proof of Theorem 3.1, which appears in Appendix A.1. The proofs of both the lower and the upper bounds rely on the analysis of Safran et al. [2022] for univariate networks that satisfy the KKT conditions. For the lower bound, we note that with high probability over the sample $(x_i)_{i=1}^m \sim \mathcal{D}_x^m$, approximately $p^5 m$ of the sampled points will be part of a sequence of 5 consecutive points $x_i < \cdots < x_{i+4}$ all labeled $-1$. Based on a lemma due to Safran et al. [2022], we show any such sequence must contain a segment $[x_j, x_{j+1}]$, $i \leq j \leq i + 3$ for which $N_{\boldsymbol{\theta}}|_{[x_j, x_{j+1}]} < 0$, contributing to the clean test error proportionally to the length of $[x_j, x_{j+1}]$. Under the probable event that most samples are not too close to one another, namely of distance $\Omega(1/m)$, we get that the total length of all such segments (and hence the error) is at least of order $\Omega(\frac{mp^5}{m}) = \Omega(p^5)$.

As to the upper bound, we observe that the number of neighboring samples with opposing labels is likely to be on order of $pm$, which implies that the data can be interpolated using a network of width of order $n^* = O(pm)$. We then invoke a result of Safran et al. [2022] that states that the class of interpolating networks that satisfy the KKT conditions has VC dimension $d_{\text{VC}} = O(n^*) = O(pm)$. Thus, a standard uniform convergence result for VC classes asserts that the test error would be of order $\frac{1}{m} \sum_{i \in [m]} \mathbb{1}\{\text{sign}(N_{\boldsymbol{\theta}})(x_i) \neq y_i\} + O(\sqrt{d_{\text{VC}}/m}) = 0 + O(\sqrt{pm/m}) = O(\sqrt{p})$.

---

[2]We remark that the additional summands are merely a matter of concentration in order to get high probability bounds. As for the expected clean test error, our proof readily shows that $cp^5 \leq \mathbb{E}_{S \sim \mathcal{D}^m}[L_{\text{cl}}(N_{\boldsymbol{\theta}})] \leq C\sqrt{p}$.

While the result above establishes the overfitting being tempered with an error scaling as $\text{poly}(p)$, we note that the range $[cp^5, C\sqrt{p}]$ is rather large. Moreover, previous works suggest that in many cases we should expect the clean test error to scale *linearly* with $p$ [Belkin et al., 2018b, Chatterji and Long, 2021, Manoj and Srebro, 2023]. Therefore it is natural to conjecture that this is also true here, at least for trained NNs that we are likely to converge to. We now turn to show such a result under a stronger assumption, where instead of any KKT point of the margin maximization problem, we consider specifically local minima. We note that several prior works have studied networks corresponding to global minima of the max-margin problem, which is of course an even stronger assumption than local minimality (e.g. Savarese et al., 2019, Boursier and Flammarion, 2023). We say that $\boldsymbol{\theta}$ is a local minimum of Problem (1) whenever there exists $r > 0$ such that if $\|\boldsymbol{\theta}' - \boldsymbol{\theta}\| < r$ and $\forall i \in [m] : y_i N_{\boldsymbol{\theta}'}(x_i) \geq 1$ then $\|\boldsymbol{\theta}\|^2 \leq \|\boldsymbol{\theta}'\|^2$.[3]

**Theorem 3.2.** *Let $d = 1$, $p \in [0, \frac{1}{2})$. Then with probability at least $1 - \delta$ over the sample $S \sim \mathcal{D}^m$, for any local minimum $\boldsymbol{\theta} = (v_j, w_j, b_j)_{j=1}^n$ of Problem (1), it holds that*

$$c\left(p - \sqrt{\frac{\log(m/\delta)}{m}}\right) \leq L_{\text{cl}}(N_{\boldsymbol{\theta}}) \leq C\left(p + \sqrt{\frac{\log(m/\delta)}{m}}\right),$$

*where $c, C > 0$ are absolute constants.*

The theorem above indeed shows that whenever the sample size is large enough, the clean test error indeed scales linearly with $p$.[4] The proof of Theorem 3.2, which appears in Appendix A.2, is based on a different analysis than that of Theorem 3.1. Broadly speaking, the proof relies on analyzing the structure of the learned prediction function, assuming it is a local minimum of the max-margin problem. More specifically, in order to obtain the lower bound we consider segments in between samples $x_{i-1} < x_i < x_{i+1}$ that are labeled $y_{i-1} = 1, y_i = -1, y_{i+1} = 1$. Note that with high probability over the sample, approximately $p(1-p)^2 = \Omega(p)$ of the probability mass lies in such segments. Our key proposition is that for a KKT point which is a local minimum, $N_{\boldsymbol{\theta}}(\cdot)$ must be linear on the segment $[x_i, x_{i+1}]$, and furthermore that $N_{\boldsymbol{\theta}}(x_i) = -1, N_{\boldsymbol{\theta}}(x_{i+1}) = 1$. Thus, assuming the points are not too unevenly spaced, it follows that a constant fraction of any such segment contributes to the clean test error, resulting in an overall error of $\Omega(p)$. In order to prove this proposition, we provide an exhaustive list of cases in which the network does *not* satisfy the assertion, and show that in each case the norm can be locally reduced - see Figure 3 in the appendix for an illustration.

For the upper bound, a similar analysis is provided for segments $[x_i, x_{i+1}]$ for which $y_i = y_{i+1} = 1$. Noting that with high probability over the sample an order of $(1-p)^2 = 1 - O(p)$ of the probability mass lies in such segments, it suffices to show that along any such segment the network is positive – implying an upper bound of $O(p)$ on the possible clean test error. Indeed, we show that along such segments, by locally minimizing parameter norm the network is incentivized to stay positive.

## 4 Benign overfitting in high dimensions

In this section we focus on the high dimensional setting. Throughout this section we assume that the dataset is sampled according to $\mathcal{D}_{\mathbf{x}} = \text{Unif}(\mathbb{S}^{d-1})$ and $d \gg m$, where $m$ is the number of samples, so that $L_{\text{cl}}(N_{\boldsymbol{\theta}}) = \Pr_{\mathbf{x} \sim \text{Unif}(\mathbb{S}^{d-1})}[N_{\boldsymbol{\theta}}(\mathbf{x}) \leq 0]$. We note that our results hold for other commonly studied distributions (see Remark 4.2). We begin by showing that if the network converges to the maximum margin solution up to some multiplicative factor, then benign overfitting occurs:

**Theorem 4.1.** *Let $\epsilon, \delta > 0$. Assume that $p \leq c_1$, $m \geq c_2 \frac{\log(1/\delta)}{p}$ and $d \geq c_3 m^2 \log\left(\frac{m}{\epsilon}\right) \log\left(\frac{m}{\delta}\right)$ for some universal constants $c_1, c_2, c_3 > 0$. Given a sample $(\mathbf{x}_i, y_i)_{i=1}^m \sim \mathcal{D}^m$, suppose $\boldsymbol{\theta} = (\mathbf{w}_j, v_j, b_j)_{j=1}^n$ is a KKT point of Problem (1) such that $\|\boldsymbol{\theta}\|^2 \leq \frac{c_4}{\sqrt{p}}\|\boldsymbol{\theta}^*\|^2$, where $\boldsymbol{\theta}^*$ is a max-margin solution of Eq. (1) and $c_4 > 0$ is some universal constant. Then, with probability at least $1 - \delta$ over $S \sim \mathcal{D}^m$ we have that $L_{\text{cl}}(N_{\boldsymbol{\theta}}) \leq \epsilon$.*

The theorem above shows that under the specified assumptions, benign overfitting occurs for any noise level $p$ which is smaller than some universal constant. Note that this result is independent of

---

[3]Note that this is merely the standard definition of local minimality of the Euclidean norm function, under the constraints imposed by interpolating the dataset.

[4]Similarly to footnote 2, our proof also characterizes the expected clean test error for any sample size as $\mathbb{E}_{S \sim \mathcal{D}^m}[L_{\text{cl}}(N_{\boldsymbol{\theta}})] = \Theta(p)$.

the number of neurons $n$, and requires $d = \tilde{\Omega}(m^2)$. The mild lower bound on $m$ is merely to ensure that the negative samples concentrate around a $\Theta(p)$ fraction of the entire dataset. Also note that the dependence on $\epsilon$ is only logarithmic, which means that in the setting we study, the clean test error decays exponentially fast with $d$.

We now turn to provide a short proof intuition, while the full proof can be found in Appendix B.1. The main crux of the proof is showing that $\sum_{j=1}^{n} v_j \sigma(b_j) = \Omega(1)$, i.e. the bias terms are the dominant factor in the output of the network, and they tend towards being positive. In order to see why this suffices to show benign overfitting, first note that $N_{\boldsymbol{\theta}}(\mathbf{x}) = \sum_{j=1}^{n} v_j \sigma(\mathbf{w}_j^\top \mathbf{x} + b_j)$ equals

$$\sum_{j=1}^{n} v_j \sigma(\mathbf{w}_j^\top \mathbf{x} + b_j) + \sum_{j=1}^{n} v_j \sigma(b_j) - \sum_{j=1}^{n} v_j \sigma(b_j) \geq \sum_{j=1}^{n} v_j \sigma(b_j) - \sum_{j=1}^{n} \left| v_j \sigma(\mathbf{w}_j^\top \mathbf{x}) \right| .$$

All the samples (including the test sample $\mathbf{x}$) are drawn independently from $\mathrm{Unif}(\mathbb{S}^{d-1})$, thus with high probability $|\mathbf{x}^\top \mathbf{x}_i| = o_d(1)$ for all $i \in [m]$. Using our assumption regarding convergence to the max-margin solution (up to some multiplicative constant), we show that the norms $\|\mathbf{w}_j\|$ are bounded by some term independent of $d$. Thus, we can bound $|\mathbf{x}^\top \mathbf{w}_j| = o_d(1)$ for every $j \in [n]$. Additionally, by the same assumption we show that both $\sum_{j=1}^{n} \|\mathbf{w}_j\|$ and $\sum_{j=1}^{n} |v_j|$ are bounded by the value of the max-margin solution (up to a multiplicative factor), which we bound by $O(\sqrt{m})$. Plugging this into the displayed equation above, and using the assumption that $d$ is sufficiently larger than $m$, we get that the $\sum_{j=1}^{n} |v_j \sigma(\mathbf{w}_j^\top \mathbf{x})|$ term is negligible, and hence the bias terms $\sum_{j=1}^{n} v_j \sigma(b_j)$ are the predominant factor in determining the output of $N_{\boldsymbol{\theta}}$.

Showing that $\sum_{j=1}^{n} v_j \sigma(b_j) = \Omega(1)$ is done in two phases. Recall the definitions of $I_+ := \{i \in [m] : y_i = 1\}$, $I_- = \{i \in [m] : y_i = -1\}$, and that by our label distribution and assumption on $p$ we have $|I_+| \approx (1-p)m \gg pm \approx |I_-|$. We first show that if the bias terms are too small, and the predictor correctly classifies the training data, then its parameter vector $\boldsymbol{\theta}$ satisfies $\|\boldsymbol{\theta}\|^2 = \Omega(|I_+|)$. Intuitively, this is because the data is nearly mutually orthogonal, so if the biases are small, the vectors $\mathbf{w}_j$ must have a large enough correlation with each different point in $I_+$ to classify it correctly. On the other hand, we explicitly construct a solution that has large bias terms, which satisfies $\|\boldsymbol{\theta}\|^2 = O(|I_-|)$. By setting $p$ small enough, we conclude that the biases cannot be too small.

**Remark 4.2** (Data distribution). *Theorem 4.1 is phrased for data sampled uniformly from a unit sphere, but can be easily extended to other isotropic distributions such as standard Gaussian and uniform over the unit ball. In fact, the only properties of the distribution we use are that $|\mathbf{x}_i^\top \mathbf{x}_j| = o_d(1)$ for every $i \neq j \in [m]$, and that $\|XX^\top\| = O(1)$ (independent of $m$) where $X$ is an $m \times d$ matrix whose rows are $\mathbf{x}_i$.*

### 4.1 Benign overfitting under KKT assumptions

In Theorem 4.1 we showed that benign overfitting occurs in high dimensional data, but under the strong assumption of convergence to the max-margin solution up to a multiplicative factor (whose value is allowed to be larger for smaller values of $p$). On the other hand, Theorem 2.1 only guarantees convergence to a KKT point of the max-margin problem (1). We note that Vardi et al. [2022] provided several examples of ReLU neural networks where there are in fact KKT points which are not a global or even local optimum of the max-margin problem.

Consequently, in this section we aim at showing a benign overfitting result by only assuming convergence to a KKT point of the max-margin problem. For such a result we use two additional assumptions, namely: (1) The output weights $v_j$ are all fixed to be $\pm 1$, while only $(\mathbf{w}_j, b_j)_{j \in [n]}$ are trained;[5] and (2) Both the noise level $p$ and the input dimension $d$ depend on $n$, the number of neurons. Our main result is the following:

**Theorem 4.3.** *Let $\epsilon, \delta > 0$. Assume that $p \leq \frac{c_1}{n^2}$, $m \geq c_2 n \log \left( \frac{1}{\delta} \right)$ and $d \geq c_3 m^4 n^4 \log \left( \frac{m}{\epsilon} \right) \log \left( \frac{m^2}{\delta} \right)$ for some universal constants $c_1, c_2, c_3 > 0$. Assume that the output weights are fixed so that $|\{j : v_j = 1\}| = |\{j : v_j = -1\}|$ while $(\mathbf{w}_j, b_j)_{j \in [n]}$ are trained, and that $N_{\boldsymbol{\theta}}$ converges to a KKT point of Problem (1). Then, with probability at least $1 - \delta$ over $S \sim \mathcal{D}^m$ we have $L_{\mathrm{cl}}(N_{\boldsymbol{\theta}}) \leq \epsilon$.*

---

[5]This assumption appears in prior works such as Cao et al. [2022], Frei et al. [2023], Xu and Gu [2023], and is primarily used to simplify the analysis.

Note that contrary to Theorem 4.1 and to the results from Section 3, here there is a dependence on $n$, the number of neurons. For $n = O(1)$ we get benign overfitting for any $p$ smaller than a universal constant. This dependence is due to a technical limitation of the proof (which we will soon explain), and it would be interesting to remove it in future work. The full proof can be found in Appendix B.2, and here we provide a short proof intuition.

The proof strategy is similar to that of Theorem 4.1, by showing that $\sum_{j=1}^{n} v_j \sigma(b_j) = \Omega(1)$, i.e. the bias terms are dominant and tend towards being positive. The reason that this suffices is proven using similar arguments to that of Theorem 4.1. The main difficulty of the proof is showing that $\sum_{j=1}^{n} v_j \sigma(b_j) = \Omega(1)$. We can write each bias term as $b_j = v_j \sum_{i \in [m]} \lambda_i y_i \sigma'_{i,j}$, where $\sigma'_{i,j} = \mathbb{1}(\mathbf{w}_j^\top \mathbf{x}_i + b_j > 1)$ is the derivative of the $j$-th ReLU neuron at the point $\mathbf{x}_i$. We first show that all the bias terms $b_j$ are positive (Lemma B.5) and that for each $i \in I_+$, there is $j \in \{1, \dots, n/2\}$ with $\sigma'_{i,j} = 1$. This means that it suffices to show that $\lambda_i$ is not too small for all $i \in I_+$, while for every $i \in I_-$, $\lambda_i$ is bounded from above.

We assume towards contradiction that the sum of the biases is small, and show it implies that $\lambda_i = \Omega\left(\frac{1}{n}\right)$ for $i \in I_+$ and $\lambda_i = O(1)$ for $i \in I_-$. Using the stationarity condition we can write for $j \in \{1, \dots, n/2\}$ that $\mathbf{w}_j = \sum_{i=1}^{m} \lambda_i y_i \sigma'_{i,j} \mathbf{x}_i$. Taking some $r \in I_+$, we have that $\mathbf{w}_j^\top \mathbf{x}_r \approx \lambda_r \sigma'_{r,j}$ since $\mathbf{x}_r$ is almost orthogonal to all other samples in the dataset. By the primal feasibility condition (Eq. (3)) $N_{\boldsymbol{\theta}}(\mathbf{x}_r) \geq 1$, hence there must be some $j \in \{1, \dots, n/2\}$ with $\sigma'_{r,j} = 1$ and with $\lambda_r$ that is larger than some constant. The other option is that the bias terms are large, which we assumed does not happen. Note that we don't know how many neurons $\mathbf{w}_j$ there are with $\sigma'_{j,r} = 1$, which means that we can only lower bound $\lambda_r$ by a term that depends on $n$.

To show that $\lambda_s$ are not too big for $s \in I_-$, we use the complementary slackness condition (Eq. (5)) which states that if $\lambda_s \neq 0$ then $N_{\boldsymbol{\theta}}(\mathbf{x}_s) = -1$. Since the sum of the biases is not large, then if there exists some $s \in I_-$ with $\lambda_s$ that is too large we would get that $N_{\boldsymbol{\theta}}(\mathbf{x}_s) < -1$ which contradicts the complementary slackness. Combining those bounds and picking $p$ to be small enough shows that $\sum_{j=1}^{n} v_j \sigma(b_j)$ cannot be too small.

## 4.2 The role of the bias and catastrophic overfitting in neural networks

In both our benign overfitting results (Theorem 4.1 and Theorem 4.3), our main proof strategy was showing that if $p$ is small enough, then the bias terms tend to be positive and dominate over the other components of the network. In this section we further study the importance of the bias terms for obtaining benign overfitting, by examining bias-less ReLU networks of the form $N_{\boldsymbol{\theta}}(\mathbf{x}) = \sum_{j=1}^{n} v_j \sigma(\mathbf{w}_j^\top \mathbf{x})$ where $\boldsymbol{\theta} = (v_j, \mathbf{w}_j)_{j \in [n]}$. We note that if the input dimension is sufficiently large and $n \geq 2$, such bias-less networks can still fit any training data. The proofs for this section can be found in Appendix C.

We begin by showing that without a bias term and with no further assumption on the network width, any solution can exhibit catastrophic behaviour:

**Proposition 4.4** (Catastrophic overfitting without bias). *Consider a bias-less network with $n = 2$ which classifies a dataset correctly. If the dataset contains at least one sample with a negative label, then $L_{\mathrm{cl}}(N_{\boldsymbol{\theta}}) \geq \frac{1}{2}$.*

While the result above is restricted to any network of width two (in particular, it holds under any assumption such as convergence to a KKT point), this already precludes the possibility of achieving either benign or tempered overfitting without further assumptions on the width of the network – in contrast to our previous results. Furthermore, we next show that in the bias-less case, the clean test error is lower bounded by some term which depends only on $n$ (which again, holds for any network so in particular under further assumptions such as convergence to a KKT point).

**Proposition 4.5** (Not benign without bias). *For any bias-less network of width $n$, $L_{\mathrm{cl}}(N_{\boldsymbol{\theta}}) \geq \frac{1}{2^n}$.*

Note that the result above does not depend on $m$ nor on $d$, thus we cannot hope to prove benign overfitting for the bias-less case unless $n$ is large, or depends somehow on both $m$ and $d$. We emphasize that Proposition 4.5 does no subsume Proposition 4.4 although they appear to be similar. Indeed, for the case of $n = 2$, Proposition 4.5 provides a bound of $\frac{1}{4}$ which does not fall under the definition of catastrophic overfitting, as opposed to Proposition 4.4. Next, we show that even for

arbitrarily large $n$ (possibly scaling with other problem parameters such as $m, d$), there are KKT points which do not exhibit benign overfitting:

**Proposition 4.6** (Benign overfitting does not follow from KKT without bias). *Consider a bias-less network, and suppose that $\mathbf{x}_j^\top \mathbf{x}_i = 0$ for every $i \neq j$. Denote $I_- := |\{i \in [m] : y_i = -1\}|$. Then there exists a KKT point $\boldsymbol{\theta}$ of Problem ([1]) with $L_{\mathrm{cl}}(N_{\boldsymbol{\theta}}) \geq \frac{1}{2} - \frac{1}{2^{|I_-|}}$.*

The result above shows that even if we're willing to consider arbitrarily large networks, there still exist KKT points which do not exhibit benign overfitting (at least without further assumptions on the data). This implies that a positive benign overfitting result in the bias-less case, if at all possible, cannot possibly apply to all KKT points – so one must make further assumptions on which KKT points to consider or resort to an alternative analysis. The assumption that $\mathbf{x}_j^\top \mathbf{x}_i = 0$ for every $i \neq j$ is made to simplify the construction of the KKT point, while we hypothesize that the proposition can be extended to the case where $\mathbf{x}_i \sim \mathrm{Unif}(\mathbb{S}^{d-1})$ (or other isotropic distributions) for large enough $d$.

## 5 Between tempered and benign overfitting for intermediate dimensions

In Section [3], we showed that tempered overfitting occurs for one-dimensional data, while in Section [4] we showed that benign overfitting occurs for high-dimensional data . In this section we complement these results by empirically studying intermediate dimensions and their relation to the number of samples. Our experiments extend an experiment from [Mallinar et al., 2022, Figure 6b] which considered the same distribution as we do, for $d = 10$ and $m$ varying from 300 to $3.6 \times 10^5$. Here we study larger values of $d$, and how the ratio between $d$ and $m$ affects the overfitting profile. We also focus on 2-layer networks (although we do show our results may extend to depth 3).

In our experiments, we trained a fully connected neural network with 2 or 3 layers, width $n$ (which will be specified later) and ReLU activations. We sampled a dataset $(\mathbf{x}_i, y_i)_{i=1}^m \sim \mathrm{Unif}(\mathbb{S}^{d-1}) \times \mathcal{D}_y$ for noise level $p \in \{0.05, 0.1, \dots, 0.5\}$. We trained the network using SGD with a constant learning rate of $0.1$, and with the logistic (cross-entropy) loss. Each experiment ran for a total of $20k$ epochs, and was repeated 10 times with different random seeds, the plots are averaged over the runs. In all of our experiments, the trained network overfitted in the sense that it correctly classified all of the (inherently noisy) training data. Furthermore, we remark that longer training times ($100k$ epochs) were also examined, and the results remained unaffected, thus we omit such plots.

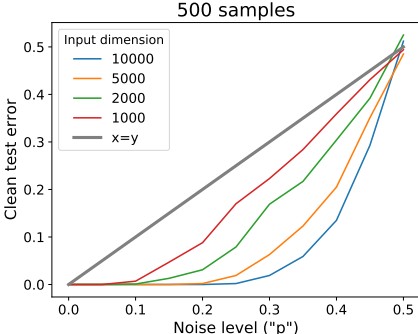
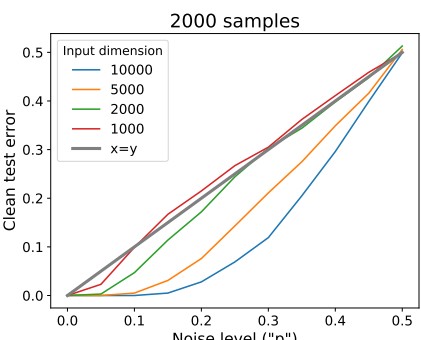

Figure 1: Training a 2-layer network with 1000 neurons on $m$ samples drawn uniformly from $\mathbb{S}^{d-1}$ for varying input dimensions $d$. Each label is equal to $-1$ with probability $p$ and $+1$ with probability $(1 - p)$. Left: $m = 500$, Right: $m = 2000$. The line corresponding to the identity function $y = x$ was added for reference. Best viewed in color.

In Figure [1] we trained a 2-layer network with 1000 neurons over different input dimensions, with $m = 500$ samples (left) and $m = 2000$ samples (right). There appears to be a gradual transition from tempered overfitting (with clean test error linear in $p$) to benign overfitting (clean test error close to 0 for any $p$), as $d$ increases compared to $m$. Our results indicate that the overfitting behavior is close to tempered/benign in wider parameter regimes than what our theoretical results formally cover ($d = 1$ vs. $d > m^2$). Additionally, benign overfitting seems to occur for lower values of $p$, whereas for $p$ closer to $0.5$ the clean test error is significantly above 0 even for the highest dimensions we examined.

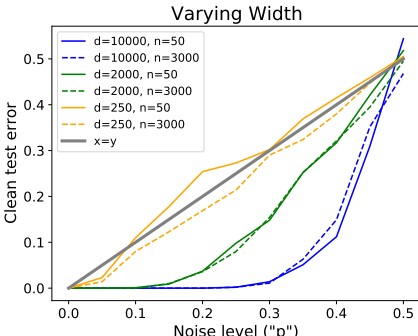 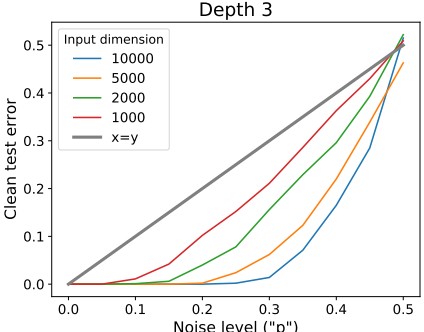

Figure 2: Left: 2-layer network with $n = 50$ and $n = 3000$ neurons and varying input dimension. Right: 3-layer network with $n = 1000$ neurons and varying input dimension. Both plots correspond to $m = 500$ samples.

We further experimented on the effects of the width and depth on the overfitting profile. In Figure 2 (left) we trained a 2-layer network on $m = 500$ samples with input dimension $d \in \{250, 2000, 10000\}$ and $n$ neurons for $n \in \{50, 3000\}$. It can be seen that even when the number of neurons vary significantly, the plots remain almost the same. It may indicate that the network width $n$ has little effect on the overfitting profile (similar to most of our theoretical results), and that the dependence on $n$ in Theorem 4.3 is an artifact of the proof technique (in accordance with our other results). In Figure 2 (right) we trained a 3-layer neural networks on $m = 500$ samples, as opposed to a 2-layer networks in all our other experiments. Noticeably, this plot looks very similar to Figure 1 (left) which may indicate that our observations generalize to deeper networks.

## 6   Discussion

In this paper, we focused on a classification setting with 2-layer ReLU neural networks that interpolate noisy training data. We showed that the implicit bias of gradient based training towards margin maximization yields tempered overfitting in a low dimensional regime, while it causes benign overfitting in high dimensions. Furthermore, we provide empirical evidence that there is a gradual transition from tempered to benign overfitting as the input dimension increases.

Our work leaves open several questions. A natural follow-up is to study data distributions for which our results do not apply. While our analysis in one dimension is readily applicable for other distributions, it is currently not clear how to extend the high dimensional analysis to other data distributions (in particular, to settings in which the Bayes-optimal predictor is not constant). Moreover, establishing formal results in any constant dimension larger than 1, or whenever the dimension scales sub-quadratically (e.g., linearly) with the number of training samples, is an interesting open problem which would likely require different proof techniques.

At a more technical level, there is the question of whether our assumptions can be relaxed. In particular, Theorem 3.2 provides a tight bound under the assumption of convergence to a local minimum of the margin maximization problem, whereas Theorem 3.1 only requires convergence to a KKT point, yet leaves a polynomial gap. Similarly, Theorem 4.1 and Theorem 4.3 require assumptions on the predictor norm or on the noise level and width, respectively.

Our work also leaves open a more thorough study of different architectures and their possible effect on the type of overfitting. In particular, we note that our results suggest that as long as the network is large enough to interpolate, the width plays a minor role in our studied setting — which we also empirically observe. On the other hand, whether this also holds for other architectures, as well as the role of depth altogether, remains unclear. Finally, it would be interesting to further study the bias-less case and assess its corresponding overfitting behavior under suitable assumptions.

## Acknowledgments and Disclosure of Funding

We thank Gal Vardi for insightful discussions regarding the implicit bias of neural networks. This research is supported in part by European Research Council (ERC) grant 754705.

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

# A   Proofs of tempered overfitting

## A.1   Proof of Theorem 3.1

Throughout the proof, given a sample $S = (x_i, y_i)_{i=1}^m \sim \mathcal{D}^m$ we denote $S_x = (x_i)_{i=1}^m$, $S_y = (y_i)_{i=1}^m$, and assume without loss of generality that $x_1 \leq x_2 \leq \cdots \leq x_m$.

**Lemma A.1.** *Denote by $E_x$ the event in which the samples $(x_i)_{i=1}^m \sim \mathcal{D}_x^m$ satisfy*

- *(maximal gap isn't too large) $d_{\max} := \max_{i \in [m-1]} (x_{i+1} - x_i) \leq \frac{\log(8(m+1)/\delta)}{m+1}$.*

- *(most gaps aren't too small) $|\{i \in [m-1] : x_{i+1} - x_i < \frac{1}{10(m+1)}\}| < \frac{m+1}{8}$.*

- *(no collisions) $\forall i \neq j \in [m] : x_i \neq x_j$.*

*Then there exists absolute $m_0 \in \mathbb{N}$ such that $\forall m \geq m_0 : \Pr_{S_x \sim \mathcal{D}_x^m}[E_x] \geq 1 - \frac{\delta}{4}$.*

*Proof.* Deferred to Appendix D. $\qquad\square$

Following the lemma above, we continue by conditioning on the probable event $E_x$, after which we will conclude the proof by the union bound. We now state a lemma due to Safran et al. [2022] which is crucial for our analysis.

**Lemma A.2** (Lemma E.6. Safran et al., 2022). *Suppose that $i < j$ are such that $y_i, y_{i+1}, \ldots, y_j = -1$. Then in the interval $[x_i, x_j]$ there are at most two points at which $N'_\theta$ increases.*

We derive the following corollary:

**Corollary A.3.** *Suppose that $y_i, y_{i+1}, \ldots, y_{i+4} = -1$. Then there exists $i \leq \ell \leq i + 3$ for which $N_\theta|_{[x_\ell, x_{\ell+1}]} < 0$.*

*Proof.* Assume towards contradiction that $y_i, \ldots, y_{i+4} = -1$, yet for any $i \leq \ell \leq i + 3$ there exists $z_\ell \in (x_\ell, x_{\ell+1})$ such that $N_\theta(z_\ell) \geq 0$. Recall that $N_\theta(x_i), \ldots, N_\theta(x_{i+3}) \leq -1$ by Eq. (3). Thus for each $i \leq \ell \leq i + 2$, looking at the segment $(z_\ell, z_{\ell+1}) \ni x_{\ell+1}$ we see that $N_\theta(z_\ell) > 0, N_\theta(x_{\ell+1}) \leq -1, N_\theta(z_\ell) > 0$. In particular, by the mean value theorem, any such segment must contain a point at which $N'_\theta$ increases. Obtaining three such points which are distinct contradicts Lemma A.2. $\qquad\square$

We assume without loss of generality that $m$ is divisible by 5 and split the index set $[m]$ into groups consecutive five indices: we let $I_1 = \{1, \ldots, 5\}, I_2 = \{6, \ldots, 10\}$ and so on up to $I_{m/5}$. Denoting by $\mu$ the (one dimensional) Lebesgue measure we get that under the event $E_x$ it holds that

$$
\mathbb{E}_{S_y \sim \mathcal{D}_y^m}\left[\Pr_{x \sim \mathcal{D}_x}[N_\theta(x) < 0]\right] = \mathbb{E}_{S_y \sim \mathcal{D}_y^m}\left[\mu(x : N_\theta(x) < 0)\right]
$$

$$
\geq \mathbb{E}_{S_y \sim \mathcal{D}_y^m}\left[\sum_{i \in [m-1]} \mu(x_{i+1} - x_i) \cdot \mathbb{1}\left\{N|_{[x_i, x_{i+1}]} < 0\right\}\right]
$$

$$
= \sum_{i \in [m-1]} \mathbb{E}_{S_y \sim \mathcal{D}_y^m}\left[(x_{i+1} - x_i) \cdot \mathbb{1}\left\{N|_{[x_i, x_{i+1}]} < 0\right\}\right]
$$

$$
= \sum_{i \in [m/5]} \sum_{l \in [I_i]} \mathbb{E}_{S_y \sim \mathcal{D}_y^m}\left[(x_{l+1} - x_l) \cdot \mathbb{1}\left\{N|_{[x_l, x_{l+1}]} < 0\right\}\right]
$$

$$
[\text{Corollary } A.3] \geq \sum_{i \in [m/5]} \sum_{l \in [I_i]} \mathbb{E}_{S_y \sim \mathcal{D}_y^m}\left[\min_{\ell \in I_i}(x_{\ell+1} - x_\ell) \cdot \mathbb{1}\left\{\forall \ell \in I_i : y_\ell = -1\right\}\right]
$$

$$
= \sum_{i \in [m/5]} \sum_{l \in [I_i]} \min_{\ell \in I_i}(x_{\ell+1} - x_\ell) \cdot \Pr[y_i, y_{i+1}, \ldots, y_{i+4} = -1]
$$

$$
\geq \widetilde{c}p^5 ,
$$

where the last inequality follows from our conditioning on $E_x$. To see why, note that the sum $\sum_{i\in[m/5]}\sum_{l\in[I_i]}\min_{\ell\in I_i}(x_{\ell+1}-x_\ell)$ is lower bounded by the sum of the $m/5$ smallest gaps, yet under $E_x$ this sum contains at least $\Omega(m)$ summands larger than $\Omega(1/m)$ — hence it is at least some constant. We conclude that as long as $E_x$ occurs we have $\mathbb{E}_{S_y\sim\mathcal{D}_y^m}\left[\Pr_{x\sim\mathcal{D}_x}[N_{\boldsymbol\theta}(x)<0]\right]\geq cp^5$. Moreover, we see by the analysis above that if a *single* label $y_l$ for some $l\in I_i\subset[m]$ is changed, this can affect $N_{\boldsymbol\theta}(x)$ only in the segment $[x_{\min_{I_i}\ell},x_{\max_{I_i}\ell}]$ which is of length at most $7d_{\max}=O\left(\frac{\log(m/\delta)}{m}\right)$. Thus we can apply McDiarmid's inequality to obtain that under $E_x$, with probability at least $1-\delta/4$ :

$$\Pr_{x\sim\mathcal{D}_x}[N_{\boldsymbol\theta}(x)<0]\geq c\left(p^5-\sqrt{\frac{\log(m/\delta)}{m}}\right)\ .$$

Overall, by union bounding over $E_x$ the inequality above holds with probability at least $1-\delta/2$, which proves the desired lower bound.

We now turn to prove the upper bound. Let $N^*(\cdot)$ be a 2-layer ReLU network of minimal width $n^*\in\mathbb{N}$ that classifies the data correctly, namely $y_iN^*(x_i)>0$ for all $i\in[m]$. Note that $n^*$ is uniquely defined by the sample while $N^*$ is not. Furthermore, $n^*$ is upper bounded by the number of neighboring samples with different labels.[6] Hence,

$$\begin{aligned}\mathbb{E}_{S_y\sim\mathcal{D}_y^m}[n^*]&\leq\mathbb{E}_{S_y\sim\mathcal{D}_y^m}[|\{i\in[m-1]:y_i\neq y_{i+1}\}|] &&(6)\\&=(m-1)\cdot\mathbb{E}_{S_y\sim\mathcal{D}_y^m}[\mathbb{1}\{y_1\neq y_2\}]\\&=(m-1)\cdot\Pr_{S_y\sim\mathcal{D}_y^m}[y_1\neq y_2]\\&=2(m-1)p(1-p)=O(pm)\ .\end{aligned}$$

We conclude that the expected width of $N^*$ (as a function of the sample) is at most $n^*=O(pm)$. By Safran et al. [2022, Theorem 4.2], this implies $N_{\boldsymbol\theta}$ belongs to a class of VC dimension $O(n^*)=O(pm)$. Thus by denoting the 0-1 loss $L^{0-1}(N_{\boldsymbol\theta})=\Pr_{(x,y)\sim\mathcal{D}}[\mathrm{sign}(N_{\boldsymbol\theta}(x))\neq y]$ and invoking a standard VC generalization bound we get

$$\mathbb{E}_{S_y\sim\mathcal{D}_y^m}\left[L^{0-1}(N_{\boldsymbol\theta})\,|\,n^*\right]\leq\underbrace{\frac{1}{m}\sum_{i=1}^m\mathbb{1}\{\mathrm{sign}(N_{\boldsymbol\theta}(x_i))\neq y_i\}}_{=0}+O\left(\sqrt{\frac{n^*}{m}}\right)=O\left(\sqrt{\frac{n^*}{m}}\right)\ .$$

By Jensen's inequality $\mathbb{E}[\sqrt{n^*}]\leq\sqrt{\mathbb{E}[n^*]}\lesssim\sqrt{pm}\implies\mathbb{E}[\sqrt{n^*/m}]\lesssim\sqrt{p}$, so by the law of total expectation

$$\mathbb{E}_{S_y\sim\mathcal{D}_y^m}[L^{0-1}(N_{\boldsymbol\theta})]=\mathbb{E}_{n^*}\left[\mathbb{E}_{S_y\sim\mathcal{D}_y^m}[L^{0-1}(N_{\boldsymbol\theta})\,|\,n^*]\right]\lesssim\mathbb{E}_{n^*}\left[\sqrt{\frac{n^*}{m}}\right]\lesssim\sqrt{p}\ . \qquad(7)$$

In order to relate the bound above to the clean test error, note that

$$\begin{aligned}L^{0-1}(N_{\boldsymbol\theta})&=\Pr_{(x,y)\sim\mathcal{D}}[\mathrm{sign}(N_{\boldsymbol\theta}(x))\neq y]\\&=(1-p)\cdot\Pr_{(x,y)\sim\mathcal{D}}[N_{\boldsymbol\theta}(x)\leq0|y=1]+p\cdot\Pr_{(x,y)\sim\mathcal{D}}[N_{\boldsymbol\theta}(x)>0|y=-1]\\&\geq\underbrace{(1-p)}_{\in[\frac{1}{2},1]}\cdot\Pr_{x\sim\mathcal{D}_x}[N_{\boldsymbol\theta}(x)\leq0]\\&\geq\frac{1}{2}\cdot\Pr_{x\sim\mathcal{D}_x}[N_{\boldsymbol\theta}(x)\leq0]\ ,\end{aligned}$$

hence

$$\mathbb{E}_{S_y\sim\mathcal{D}_y^m}\left[\Pr_{x\sim\mathcal{D}_x}[N_{\boldsymbol\theta}(x)\leq0]\right]\leq\mathbb{E}_{S_y\sim\mathcal{D}_y^m}\left[2L^{0-1}(N_{\boldsymbol\theta})\right]\overset{\mathrm{Eq.\ (7)}}{\lesssim}\sqrt{p}\ .$$

---

[6]This can be seen by considering a network representing the linear spline of the data, for which it suffices to set a neuron for adjacent samples with alternating signs.

As in our argument for the lower bound, we now note by Eq. (6) that flipping a single label $y_l$ for some $l \in [m]$ changes $n^*$ by at most 1, hence changing the test error by at most $O(1/m)$. Therefore we can apply McDiarmid's inequality and see that under the event $E_x$, with probability at least $1 - \delta/4$:

$$\Pr_{x \sim \mathcal{D}_x} [N_{\boldsymbol{\theta}}(x) \leq 0] \leq C \left( \sqrt{p} + \sqrt{\frac{\log(1/\delta)}{m}} \right) .$$

Overall, by union bounding over $E_x$ the inequality above holds with probability at least $1 - \delta/2$, which proves the upper bound and finishes the proof.

## A.2 Proof of Theorem 3.2

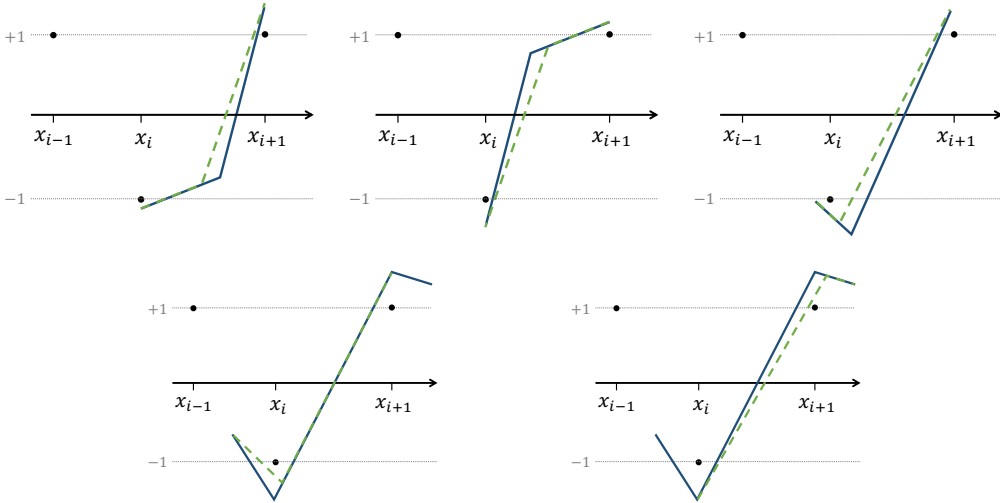

Figure 3: Illustration of the proof of Theorem 3.2 in case there is a single non-linearity along $[x_i, x_{i+1}]$. If the network is not linear along $[x_i, x_{i+1}]$, one of the cases illustrated in the top row (in blue) must occur. In each case, the dashed green perturbation classifies correctly by altering exactly two neurons while reducing the parameter norm. Moreover, if the network is linear along $[x_i, x_{i+1}]$, yet $N_{\boldsymbol{\theta}}(x_i) < -1$ or $N_{\boldsymbol{\theta}}(x_{i+1}) > 1$, one of the cases illustrated in the bottom row must occur. In either case, the dashed green perturbation classifies correctly by altering exactly two neurons while reducing the parameter norm.

Throughout the proof, given a sample $S = (x_i, y_i)_{i=1}^m \sim \mathcal{D}^m$ we denote $S_x = (x_i)_{i=1}^m$, $S_y = (y_i)_{i=1}^m$, and assume without loss of generality that $x_1 \leq x_2 \leq \cdots \leq x_m$. Denote by $E_x$ the event in which the samples $(x_i)_{i=1}^m \sim \mathcal{D}_x^m$ satisfy

$$d_{\max} := \max_{i \in [m-1]} (x_{i+1} - x_i) \leq \frac{\log(8(m+1)/\delta)}{m+1} ,$$

and recall that by Lemma A.1 we have $\Pr_{S_x \sim \mathcal{D}_x^m}[E_x] \geq 1 - \frac{\delta}{4}$ for $m$ larger than an absolute constant. Therefore from here on throughout the proof we condition on $E_x$, after which we can conclude using the union bound.

For any point $x \in (0, 1)$, denote by $i_x$ the maximal index $i \in [m]$ such that $x_i \leq x$. Let $A_x$ denote the event in which $y_{i_x - 1} = 1$, $y_{i_x} = -1$ and $y_{i_x + 1} = 1$. Note that for any $x$ we have $\Pr_{S_y \sim \mathcal{D}_y^m}[A_x] = p(1-p)^2 \geq \frac{1}{4}p$, so we get

$$\mathbb{E}_{S_y \sim \mathcal{D}_y^m} \left[ \Pr_{x \sim \mathcal{D}_x} [N_{\boldsymbol{\theta}}(x) < 0] \right] = \mathbb{E}_{S_y \sim \mathcal{D}_y^m, x \sim \mathcal{D}_x} [\mathbb{1}\{N_{\boldsymbol{\theta}}(x) < 0\}]$$

$$\geq \mathbb{E}_{S_y \sim \mathcal{D}_y^m, x \sim \mathcal{D}_x} [\mathbb{1}\{N_{\boldsymbol{\theta}}(x) < 0\} | A_x] \cdot \Pr_{S_y \sim \mathcal{D}_y^m, x \sim \mathcal{D}_x} [A_x]$$

$$\geq \frac{p}{4} \cdot \mathbb{E}_{S_y \sim \mathcal{D}_y^m} \left[ \Pr_{x \sim \mathcal{D}_x} [N_{\boldsymbol{\theta}}(x) < 0 | A_x] \right] . \tag{8}$$

We aim to show that $\Pr_{x \sim \mathcal{D}_x} [N_{\boldsymbol{\theta}}(x) < 0 | A_x] = \Omega(1)$. In order to do so, note that conditioning the uniform measure $\mathcal{D}_x = \mathrm{Unif}([0,1])$ on the event $A_x$ results in a uniform measure over the (union of) segments between $-1$ labeled samples to $+1$ labeled samples that also satisfy the additional property that the neighboring sample on their left is labeled $+1$. We will show that if $\boldsymbol{\theta}$ is a local minimum of Problem (1), then along any such segment $N_{\boldsymbol{\theta}}(x)$ is linear from $-1$ to $+1$:

**Proposition A.4.** *Let $i \in [m-1]$ be such that $x_{i-1} < x_i < x_{i+1}$ with $y_{i-1} = 1$, $y_i = -1$ and $y_{i+1} = 1$. If $\boldsymbol{\theta}$ is a local minimum of Problem (1), it holds that $N_{\boldsymbol{\theta}}(x_i) = -1$, $N_{\boldsymbol{\theta}}(x_{i+1}) = 1$ and $N_{\boldsymbol{\theta}}(\cdot)$ is linear over $(x_i, x_{i+1})$.*

In particular, the proposition above shows that

$$\Pr_{x \sim \mathcal{D}_x} [N_{\boldsymbol{\theta}}(x) < 0 | A_x] = \frac{1}{2} ,$$

which plugged into Eq. (8) gives

$$\mathbb{E}_{S_y \sim \mathcal{D}_y^m} \left[ \Pr_{x \sim \mathcal{D}_x} [N_{\boldsymbol{\theta}}(x) < 0] \right] \geq \frac{p}{8} .$$

Moreover, noting that any local minimum is in particular a KKT point, we saw in the proof of Theorem 3.1 that under the event $E_x$, changing a single label $y_l$, $l \in [m]$ can change the test error by at most $O(d_{\max}) = O(\log(m/\delta)/m)$. By McDiarmid's inequality this finishes the proof of the desired lower bound.

As for the upper bound, for $x \in (0,1)$ we denote by $B_x$ the event in which $y_{i_x} = 1 = y_{i_x+1} = 1$ — namely, $x$ is between two positively labeled samples. Note that for any $x$ we have $\Pr_{S_y \sim \mathcal{D}_y^m}[B_x] = (1-p)^2 \geq 1 - 2p \implies \Pr_{S_y \sim \mathcal{D}_y^m}[B_x^c] \leq 2p$. We will show that if $\boldsymbol{\theta}$ is a local minimum of the margin maximization problem, then $B_x$ implies that $N_{\boldsymbol{\theta}}(x) \geq 0$.

**Proposition A.5.** *Let $i \in [m-1]$ be such that $y_i = y_{i+1} = 1$, and let $x \in [x_i, x_{i+1}]$. If $\boldsymbol{\theta}$ is a local minimum of Problem (1), it holds that $N_{\boldsymbol{\theta}}(x) \geq 0$.*

In particular, the proposition above shows that

$$\mathbb{E}_{S_y \sim \mathcal{D}_y^m, x \sim \mathcal{D}_x} [\mathbb{1}\{N_{\boldsymbol{\theta}}(x) < 0\} | B_x] = 0 ,$$

so we get

$$\mathbb{E}_{S_y \sim \mathcal{D}_y^m} \left[ \Pr_{x \sim \mathcal{D}_x} [N_{\boldsymbol{\theta}}(x) < 0] \right] = \mathbb{E}_{S_y \sim \mathcal{D}_y^m, x \sim \mathcal{D}_x} [\mathbb{1}\{N_{\boldsymbol{\theta}}(x) < 0\}]$$

$$= \mathbb{E}_{S_y \sim \mathcal{D}_y^m, x \sim \mathcal{D}_x} [\mathbb{1}\{N_{\boldsymbol{\theta}}(x) < 0\} | B_x] \cdot \Pr_{S_y \sim \mathcal{D}_y^m, x \sim \mathcal{D}_x} [B_x]$$

$$+ \mathbb{E}_{S_y \sim \mathcal{D}_y^m, x \sim \mathcal{D}_x} [\mathbb{1}\{N_{\boldsymbol{\theta}}(x) < 0\} | B_x^c] \cdot \Pr_{S_y \sim \mathcal{D}_y^m, x \sim \mathcal{D}_x} [B_x^c]$$

$$\leq 0 + 1 \cdot 2p = 2p .$$

As in the lower bound proof, recalling that any local minimum is in particular a KKT point, we saw in the proof of Theorem 3.1 that under the event $E_x$, changing a single label $y_l$, $l \in [m]$ can change the test error by at most $O(d_{\max}) = O(\log(m/\delta)/m)$. Hence applying McDiarmid's inequality proves the upper bound thus finishing the proof.

*Proof of Proposition A.4.* Throughout the proof we fix $i \in [m]$ for which the conditions described in the proposition hold, and we assume without loss of generality that the neurons are ordered

with respect to their activation point: $-\frac{b_1}{w_1} \leq -\frac{b_2}{w_2} \leq \cdots \leq -\frac{b_n}{w_n}$. Moreover, we may assume without loss of generality that $w_1, \ldots, w_n \geq 0$. Indeed, note that Eq. (1) is invariant under the transformation $v_j \leftarrow -v_j, w_j \leftarrow -w_j, b_j \leftarrow -b_j$ which does not affect neither the parameter norm nor the parameterized network in function space. Hence, any local minimum of Eq. (1) corresponds to a local minimum with $w_1, \ldots, w_n \geq 0$. Lastly, we will make frequent use of the following simple observation. For differentiable $x$ we have

$$N'_{\boldsymbol{\theta}}(x) = \sum_{j=1}^{n} v_j \cdot \underbrace{w_j \mathbb{1}\{w_j x + b_j \geq 0\}}_{\geq 0}, \tag{9}$$

so if $N'_{\boldsymbol{\theta}}(z_1) < N'_{\boldsymbol{\theta}}(z_2)$ for some $z_1 < z_2$, then there must exist $j \in [n]$ with $v_j > 0$ and $z_1 < -\frac{b_j}{w_j} < z_2$. Similarly, $N'_{\boldsymbol{\theta}}(z_1) > N'_{\boldsymbol{\theta}}(z_2)$ implies that there exists $j \in [n]$ with $v_j < 0$, $z_1 < -\frac{b_j}{w_j} < z_2$.

We split the proof of Proposition A.4 into two lemmas.

**Lemma A.6.** $N_{\boldsymbol{\theta}}(\cdot)$ *is linear over* $(x_i, x_{i+1})$.

*Proof.* Recall that $N_{\boldsymbol{\theta}}(x_i) \leq -1$ and $N_{\boldsymbol{\theta}}(x_{i+1}) \geq 1$, so $N_{\boldsymbol{\theta}}$ increases along the segment $(x_i, x_{i+1})$. Thus, $N_{\boldsymbol{\theta}}(\cdot)$ is *not* linear over $(x_i, x_{i+1})$ only if (at least) one of the following occur: (1) There exist $z_1, z_2 \in (x_i, x_{i+1})$ such that $z_1 < z_2$ and $0 < N'_{\boldsymbol{\theta}}(z_1) < N'_{\boldsymbol{\theta}}(z_2)$; (2) there exist $z_1, z_2 \in (x_i, x_{i+1})$ such that $z_1 < z_2$, $N'_{\boldsymbol{\theta}}(z_1) > 0$ and $N'_{\boldsymbol{\theta}}(z_1) > N'_{\boldsymbol{\theta}}(z_2)$; or (3) there exists a single $z \in (x_i, x_{i+1})$ at which $N_{\boldsymbol{\theta}}$ is non-differentiable, such that $N'_{\boldsymbol{\theta}}|_{(x_i,z)} \leq 0$ and $N'_{\boldsymbol{\theta}}|_{(z,x_{i+1})} > 0$. We will show either of these contradict the assumption that $\boldsymbol{\theta}$ is a local optimum of the margin maximization problem.

**Case (1).** The assumption on $z_1, z_2$ implies that there exists $j_1 \in [n]$ such that $-\frac{b_{j_1}}{w_{j_1}} \in (z_1, z_2)$ and $v_{j_1} > 0$. Let $j_2 > j_1$ be the minimal index $j \in \{j_1 + 1, \ldots, n\}$ for which $v_j < 0$.[7] For some small $\delta > 0$, consider the perturbed network

$$N_{\boldsymbol{\theta}_\delta}(x) := \sum_{j \in [n] \setminus \{j_1, j_2\}} v_j \sigma(w_j \cdot x + b_j) \tag{10}$$
$$+ (1 - \delta) v_{j_1} \sigma \left( w_{j_1} \cdot x + \left( b_{j_1} - \frac{\delta}{1 - \delta} \left( \frac{w_{j_1} b_{j_2}}{w_{j_2}} - b_{j_1} \right) \right) \right)$$
$$+ \left( 1 + \delta \frac{v_{j_1} w_{j_1}}{v_{j_2} w_{j_2}} \right) v_{j_2} \sigma(w_{j_2} \cdot x + b_{j_2}) .$$

It is clear that $\|\boldsymbol{\theta} - \boldsymbol{\theta}_\delta\| \overset{\delta \to 0}{\longrightarrow} 0$, and we will show that for small enough $\delta$ the network above still satisfies the margin condition, yet has smaller parameter norm. To see why the margin condition is not violated for small enough $\delta$, notice that $N_{\boldsymbol{\theta}_\delta}(x) = N_{\boldsymbol{\theta}}(x)$ for all $x \notin (x_i, -\frac{b_{j_2}}{w_{j_2}})$ so in particular $N_{\boldsymbol{\theta}_\delta}(x_l) = N_{\boldsymbol{\theta}}(x_l)$ for all $l \in [i]$, as well as for all $x_l \geq -\frac{b_{j_2}}{w_{j_2}}$. Furthermore, by minimality of $j_2$ we note that there cannot exist $y_k = -1$ for $k$ such that $x_k \in (-\frac{b_{j_1}}{w_{j_1}}, -\frac{b_{j_2}}{w_{j_2}})$. A direct computation gives that $N_{\boldsymbol{\theta}_\delta} \geq N_{\boldsymbol{\theta}}$ along this segment, so overall the margin condition is indeed satisfied for the

---

[7]We can assume that such $j_2$ exists, by otherwise discarding $j_2$ in the rest of the proof which would work verbatim. Notably, the only case in which there does not exist such $j_2$ is when $x_i$ is the last sample to be labeled $-1$.

entire sample. As to the parameter norm, we have

$$\|\boldsymbol{\theta}_\delta\|^2 = \sum_{j\in[n]\setminus\{j_1,j_2\}} (v_j^2 + w_j^2 + b_j^2) + (1-\delta)^2 v_{j_1}^2 + w_{j_1}^2 + \left(b_{j_1} - \frac{\delta}{1-\delta}\left(\frac{w_{j_1}b_{j_2}}{w_{j_2}} - b_{j_1}\right)\right)^2$$

$$+ \left(1 + \delta\frac{v_{j_1}w_{j_1}}{v_{j_2}w_{j_2}}\right)^2 v_{j_2}^2 + w_{j_2}^2 + b_{j_2}^2$$

$$= \sum_{j\in[n]\setminus\{j_1,j_2\}} (v_j^2 + w_j^2 + b_j^2) + (1-2\delta)v_{j_1}^2 + w_{j_1}^2 + b_{j_1}^2 - \frac{2\delta}{1-\delta}b_{j_1}\left(\frac{w_{j_1}b_{j_2}}{w_{j_2}} - b_{j_1}\right)$$

$$+ \left(1 + 2\delta\frac{v_{j_1}w_{j_1}}{v_{j_2}w_{j_2}}\right) v_{j_2}^2 + w_{j_2}^2 + b_{j_2}^2 + O(\delta^2)$$

$$= \sum_{j\in[n]} (v_j^2 + w_j^2 + b_j^2) - 2\delta\left(v_{j_1}^2 + \frac{b_{j_1}}{1-\delta}\left(\frac{w_{j_1}b_{j_2}}{w_{j_2}} - b_{j_1}\right) - \frac{v_{j_1}w_{j_1}}{v_{j_2}w_{j_2}}\right) + O(\delta^2) .$$

Thus,

$$\|\boldsymbol{\theta}\|^2 - \|\boldsymbol{\theta}_\delta\|^2 = 2\delta\left(v_{j_1}^2 - \frac{v_{j_1}w_{j_1}v_{j_2}}{w_{j_2}} - \frac{b_{j_1}}{1-\delta}\left(b_{j_1} - \frac{w_{j_1}b_{j_2}}{w_{j_2}}\right)\right) + O(\delta^2)$$

$$\implies \|\boldsymbol{\theta}\|^2 = \|\boldsymbol{\theta}_\delta\|^2 + 2\delta\left(v_{j_1}^2 - \frac{v_{j_1}w_{j_1}v_{j_2}}{w_{j_2}} - \frac{b_{j_1}}{1-\delta}\left(b_{j_1} - \frac{w_{j_1}b_{j_2}}{w_{j_2}}\right)\right) + O(\delta^2) . \quad (11)$$

By construction, $v_{j_1} > 0$ and $v_{j_2} < 0$ hence $-\frac{v_{j_1}w_{j_1}v_{j_2}}{w_{j_2}} > 0$. Moreover, $-\frac{b_{j_1}}{w_{j_1}} > 0$ and $w_{j_1} > 0$ so $-\frac{b_{j_1}}{1-\delta} > 0$ for $\delta < 1$. Lastly, recall that $j_2 > j_1 \implies -\frac{b_{j_2}}{w_{j_2}} \geq -\frac{b_{j_1}}{w_{j_1}} \implies b_{j_1} - \frac{w_{j_1}b_{j_2}}{w_{j_2}} \geq 0$. Overall, plugging these into Eq. (11) shows that $\|\boldsymbol{\theta}\|^2 > \|\boldsymbol{\theta}_\delta\|^2$ for small enough $\delta$, contradicting the assumption the $\boldsymbol{\theta}$ is a local minimum.

**Case (2).** Since $N_{\boldsymbol{\theta}}(x_{i-1}) \geq 1$ and $N_{\boldsymbol{\theta}}(x_i) \leq -1$, $N_{\boldsymbol{\theta}}'$ must be negative somewhere along the segment $(x_{i-1}, x_i)$. Yet, $N_{\boldsymbol{\theta}}'(z_1) > 0$ so there must exist $j_1 \in [n]$ such that $-\frac{b_{j_1}}{w_{j_1}} \in (x_{i-1}, z_1)$ and $v_{j_1} > 0$. Moreover, the assumption on $z_1, z_2$ implies that there exists $j_2 \in [n]$ such that $-\frac{b_{j_2}}{w_{j_2}} \in (z_1, z_2)$ and $v_{j_2} < 0$. For some small $\delta > 0$, consider the perturbed network

$$N_{\boldsymbol{\theta}_\delta}(x) := \sum_{j\in[n]\setminus\{j_1,j_2\}} v_j\sigma\left(w_j \cdot x + b_j\right) \quad (12)$$

$$+ (1-\delta)v_{j_1}\sigma\left(w_{j_1} \cdot x + b_{j_1}\right) + v_{j_2}\sigma\left(\left(w_{j_2} + \frac{\delta v_{j_1}w_{j_1}}{v_{j_2}}\right) \cdot x + \left(b_{j_2} + \frac{\delta v_{j_1}b_{j_1}}{v_{j_2}}\right)\right) .$$

It is clear that $\|\boldsymbol{\theta} - \boldsymbol{\theta}_\delta\| \xrightarrow{\delta\to0} 0$, and we will show that for small enough $\delta$ the network above still satisfies the margin condition, yet has smaller parameter norm. To see why the margin condition is not violated for small enough $\delta$, notice that $N_{\boldsymbol{\theta}_\delta}(x) = N_{\boldsymbol{\theta}}(x)$ for all $x \notin (x_{i-1}, x_{i+1})$ so in particular $N_{\boldsymbol{\theta}_\delta}(x_l) = N_{\boldsymbol{\theta}}(x_l)$ for all $l \neq i$. Furthermore, a direct computation yields $N_{\boldsymbol{\theta}_\delta}(x_i) < N_{\boldsymbol{\theta}}(x_i) \leq 1$. As to the parameter norm, by a similar computation to that leading up to Eq. (11) we get that

$$\|\boldsymbol{\theta}\|^2 = \|\boldsymbol{\theta}_\delta\|^2 + 2\delta\left(v_{j_1}^2 - \frac{v_{j_1}w_{j_1}w_{j_2}}{v_{j_2}} - \frac{v_{j_1}b_{j_1}b_{j_2}}{v_{j_2}}\right) + O(\delta^2) . \quad (13)$$

By construction, $v_{j_1}w_{j_1} > 0$ and $v_{j_2}w_{j_2} < 0$ hence $-\frac{v_{j_1}w_{j_1}w_{j_2}}{v_{j_2}} > 0$. Moreover, $-\frac{b_{j_1}}{w_{j_1}} > 0$ and $-\frac{b_{j_2}}{w_{j_2}} > 0$ so we also have $-\frac{v_{j_1}b_{j_1}b_{j_2}}{v_{j_2}} > 0$. Hence, Eq. (13) shows that $\|\boldsymbol{\theta}\|^2 > \|\boldsymbol{\theta}_\delta\|^2$ for small enough $\delta$, contradicting the assumption the $\boldsymbol{\theta}$ is a local minimum.

**Case (3).** The assumption implies that there exists $j_1 \in [n]$ such that $-\frac{b_{j_1}}{w_{j_1}} = z \in (x_i, x_{i+1})$, $v_{j_1} > 0$ and $-\frac{b_{j_1+1}}{w_{j_1+1}} \geq x_{i+1}$. Let $j_2 > j_1$ be the minimal index $j \in \{j_1 + 1, \ldots, n\}$ such that $v_j < 0$ (see Footnote 7). Consider the perturbed netowrk as in Eq. (10) and continue the proof as in Case (1) verbatim.

$\square$

**Lemma A.7.** $N_{\boldsymbol{\theta}}(x_i) = -1$ and $N_{\boldsymbol{\theta}}(x_{i+1}) = 1$.

*Proof of Lemma A.7.* Recall that $N_{\boldsymbol{\theta}}(x_i) \leq -1$, so assume towards contradiction that $N_{\boldsymbol{\theta}}(x_i) < -1$. Note that since $N_{\boldsymbol{\theta}}(x_{i-1}) \geq 1$, $N_{\boldsymbol{\theta}}(\cdot)$ must decrease along $(x_{i-1}, x_i)$, so there must exist $j \in [n]$ with $v_j < 0$ and $-\frac{b_j}{w_j} < x_i$. Denote by $j_1 \in [n]$ the maximal such index. Similarly, since $N_{\boldsymbol{\theta}}(x_{i+1}) \geq 1$ there must exist $j_2 \in [n]$ with $v_{j_2} > 0$ and $x_{i-1} < -\frac{b_{j_2}}{w_{j_2}} < x_{i+1}$. Consider the perturbed network

$$
N_{\boldsymbol{\theta}_\delta}(x) := \sum_{j \in [n] \setminus \{j_1, j_2\}} v_j \sigma\left(w_j \cdot x + b_j\right) \tag{14}
$$
$$
+ (1-\delta)v_{j_1}\sigma\left(w_{j_1} \cdot x + b_{j_1}\right) + v_{j_2}\sigma\left(\left(w_{j_2} + \frac{\delta v_{j_1} w_{j_1}}{v_{j_2}}\right) \cdot x + \left(b_{j_2} + \frac{\delta v_{j_1} b_{j_1}}{v_{j_2}}\right)\right)
$$

for some small $\delta > 0$. It is clear that $\|\boldsymbol{\theta} - \boldsymbol{\theta}_\delta\| \xrightarrow{\delta \to 0} 0$, and we will show that for small enough $\delta$ the network above still satisfies the margin condition, yet has smaller parameter norm. To see why the margin condition is not violated for small enough $\delta$, first notice that $N_{\boldsymbol{\theta}_\delta}(x) = N_{\boldsymbol{\theta}}(x)$ for all $x \notin (x_{i-1}, x_{i+1})$. Furthermore, $N_{\boldsymbol{\theta}_\delta}(x_i)$ is continuous with respect to $\delta$ so $N_{\boldsymbol{\theta}}(x_i) < -1$ implies that $N_{\boldsymbol{\theta}_\delta}(x_i) < -1$ for small enough $\delta$. As to the parameter norm, by a similar computation to that leading up to Eq. (11) we get that

$$
\|\boldsymbol{\theta}\|^2 = \|\boldsymbol{\theta}_\delta\|^2 + 2\delta\left(v_{j_1}^2 - \frac{v_{j_1} w_{j_1} w_{j_2}}{v_{j_2}} - \frac{v_{j_1} b_{j_1} b_{j_2}}{v_{j_2}}\right) + O(\delta^2). \tag{15}
$$

By construction, $v_{j_1} < 0$ and $v_{j_2} > 0$ hence $-\frac{v_{j_1} w_{j_1} w_{j_2}}{v_{j_2}} > 0$. Moreover, $-\frac{b_{j_1}}{w_{j_1}}$ and $-\frac{b_{j_2}}{w_{j_2}} > 0$ so we also have $-\frac{v_{j_1} b_{j_1} b_{j_2}}{v_{j_2}} > 0$. Hence, Eq. (15) shows that $\|\boldsymbol{\theta}\|^2 > \|\boldsymbol{\theta}_\delta\|^2$ for small enough $\delta$, contradicting the assumption the $\boldsymbol{\theta}$ is a local minimum.

Having proved that $N_{\boldsymbol{\theta}}(x_i) = -1$, we turn to prove that $N_{\boldsymbol{\theta}}(x_{i+1}) = 1$. Knowing that $N_{\boldsymbol{\theta}}(x_{i+1}) \geq 1$, we assume towards contradiction that $N_{\boldsymbol{\theta}}(x_{i+1}) > 1$. Recalling that $N_{\boldsymbol{\theta}}(x_{i-1}) \geq 1$, $N_{\boldsymbol{\theta}}(x_i) \leq 1$ and that $N_{\boldsymbol{\theta}}(\cdot)$ is linear along $(x_i, x_{i+1})$ due to Lemma A.6, we conclude that there must exist $j_1 \in [n]$ such that $v_{j_1} > 0$ and $x_{i-1} < -\frac{b_{j_1}}{w_{j_1}} \leq x_i$. Denote by $j_2 > j_1$ the minimal index such that $v_{j_2} < 0$. Consider once again the perturbed network in Eq. (14) (only now $j_1, j_2$ are different, as we just described). The same argument as in the previous part of the proof shows that for small enough $\delta$ the network above still satisfies the margin condition, while Eq. (15) once again implies that $\|\boldsymbol{\theta}\|^2 > \|\boldsymbol{\theta}_\delta\|^2$ for small enough $\delta$ – contradicting the assumption the $\boldsymbol{\theta}$ is a local minimum. $\qquad \square$

Overall, combining Lemma A.6 and Lemma A.7 finishes the proof of Proposition A.4. $\qquad \square$

*Proof of Proposition A.5.* The proof is essentially the same as Case (1) in the proof of Proposition A.4.

Throughout the proof we fix $i \in [m-1]$ for which the conditions described in the proposition hold, and we assume without loss of generality that the neurons are ordered with respect to their activation point: $-\frac{b_1}{w_1} \leq -\frac{b_2}{w_2} \leq \cdots \leq -\frac{b_n}{w_n}$. As in the proof of Proposition A.4, we may assume without loss of generality that $w_1, \ldots, w_n \geq 0$. Moreover, as explained there as a consequence of Eq. (9), we observe that $N'_{\boldsymbol{\theta}}(z_1) < N'_{\boldsymbol{\theta}}(z_2)$ for some $z_1 < z_2$ implies the existence of $j_1 \in [n]$ with $v_{j_1} > 0$ and $z_1 < -\frac{b_{j_1}}{w_{j_1}} < z_2$. Similarly, $N'_{\boldsymbol{\theta}}(z_1) > N'_{\boldsymbol{\theta}}(z_2)$ implies that there exists $j_2 \in [n]$ with $v_{j_2} < 0$, $z_1 < -\frac{b_{j_2}}{w_{j_2}} < z_2$. With this choice of $j_1, j_2$, the proof continues as in Case (1) in the proof of Proposition A.4 verbatim.

$\qquad \square$

# B Proofs of benign overfitting

## B.1 Proof of Theorem 4.1

We sample a dataset $\mathbf{x}_1, \ldots, \mathbf{x}_m \sim \text{Unif}(\mathbb{S}^{d-1})$, and labels $y_1, \ldots, y_m \sim \mathcal{D}_y$, for $p \leq c_1$ for some universal constant $c_1$. We first prove that the following properties holds with probability $> 1 - \delta$:

1. For every $i, j \in [m]$, $|\langle \mathbf{x}_i, \mathbf{x}_j \rangle| \leq \sqrt{\frac{2 \log\left(\frac{3m^2}{\delta}\right)}{d}}$

2. $\|XX^\top\| \leq C$ where $C$ is some universal constant, and $X$ is a matrix whose rows are equal to $\mathbf{x}_i$.

3. $|I_-| \leq \frac{3pm}{2}$.

**Lemma B.1.** *Let $\delta > 0$, assume that we sample $\mathbf{x}_1, \ldots \mathbf{x}_m \sim \text{Unif}(\mathbb{S}^{d-1})$, and $y_1, \ldots, y_m \sim \mathcal{D}_y$ for $p \leq c_1$, and that $m > c_2 \frac{\log\left(\frac{1}{\delta}\right)}{p}$, for some universal constant $c_1, c_2 > 0$. Then, with probability $> 1 - \delta$ properties 1, 2, 3 holds.*

*Proof.* By Lemma D.1 we have that $\Pr\left[|\mathbf{x}_i^\top \mathbf{x}_j| \geq \sqrt{\frac{2 \log\left(\frac{1}{\delta'}\right)}{d}}\right] \leq \delta'$. Take $\delta' = \frac{\delta}{3m^2}$, and use union bound over all $i, j \in [m]$ with $i \neq j$. This shows that for every $i \neq j$ we have:

$$\Pr\left[|\mathbf{x}_i^\top \mathbf{x}_j| \geq \sqrt{\frac{2 \log\left(\frac{3m^2}{\delta}\right)}{d}}\right] \leq \frac{\delta}{3} .$$

This proves Property 1. Next, set $X$ to be the matrix whose rows are equal to $\mathbf{x}_i$. By Lemma D.2 there is a constant $c' > 0$ such that:

$$\Pr\left[\|XX^\top - I\| \geq \frac{c'}{d}\left(\sqrt{\frac{d + \log\left(\frac{3}{\delta}\right)}{m}} + \frac{d + \log\left(\frac{3}{\delta}\right)}{m}\right)\right] \leq \frac{\delta}{3} . \tag{16}$$

We can also bound $\|XX^\top\| \leq \|I\| + \|XX^\top - I\| \leq 1 + \|XX^\top - I\|$. Combining both bounds and using the assumption that $m \geq \log\left(\frac{3}{\delta}\right)$ we get that there is a universal constant $C > 0$ such that $\Pr\left[\|XX^\top\| > C\right] \leq \frac{\delta}{3}$. This proves Property 2.

Finally, using Bernstein's inequality over the choice of the labels $y_i$ have that:

$$\Pr\left[|I_-| \geq \frac{3pm}{2}\right] \leq \exp\left(-\frac{p^2 m^2/4}{mp(1-p) + mp/6}\right) \leq \exp\left(-\frac{pm}{5}\right) ,$$

where we used that $p \leq 1$. Hence, if $m \geq c_2 \frac{\log\left(\frac{1}{\delta}\right)}{p}$ for some universal constant $c_2 \geq 0$, then $|I_-| \leq \frac{3pm}{2}$. Applying union over those three arguments proves the lemma. $\square$

From now on we condition on the event that properties 1, 2, 3 hold, and our bounds will depend on the probability of this event.

In the following lemma we show that if $\boldsymbol{\theta}$ converges to a solution of the max margin problem and the bias terms are relatively small, then the norm of $\boldsymbol{\theta}$ is relatively large:

**Lemma B.2.** *Assume that $m \geq c_2 \log\left(\frac{3}{\delta}\right)$ and that $\boldsymbol{\theta} = (\mathbf{w}_j, v_j, b_j)_{j=1}^n$ is a solution to the max margin problem of Eq. (1) with $\sum_{j=1}^n v_j \sigma(b_j) \leq \frac{1}{2}$. Then, $\sum_{j=1}^n \|\mathbf{w}_j\|^2 + v_j^2 + b_j^2 \geq C\sqrt{|I_+|}$ where $C$ is some universal constant.*

*Proof.* Take $i \in [m]$, then we have that $N_{\boldsymbol{\theta}}(\mathbf{x}_i) \geq 1$. By our assumption, this implies that:

$$\frac{1}{2} \leq N_{\boldsymbol{\theta}}(\mathbf{x}_i) - \sum_{j=1}^{n} v_j \sigma(b_j)$$

$$= \sum_{j=1}^{n} v_j \sigma(\mathbf{w}_j^\top \mathbf{x}_i + b_j) - \sum_{j=1}^{n} v_j \sigma(b_j)$$

$$\leq \sum_{j=1}^{n} |v_j| \cdot \left| \sigma(\mathbf{w}_j^\top \mathbf{x}_i + b_j) - \sigma(b_j) \right|$$

$$\leq \sum_{j=1}^{n} |v_j| \cdot |\mathbf{w}_j^\top \mathbf{x}_i|$$

$$\leq \sqrt{\sum_{j=1}^{n} v_j^2} \sqrt{\sum_{j=1}^{n} (\mathbf{w}_j^\top \mathbf{x}_i)^2} \; ,$$

where in the last inequality we used Cauchy-Schwartz. Denote by $S := \sum_{j=1}^{n} \|\mathbf{w}_j\|^2 + v_j^2 + b_j^2$. Combining the above and that $\sqrt{\sum_{j=1}^{n} v_j^2} \leq \sqrt{S}$ we get:

$$\sum_{j=1}^{n} (\mathbf{w}_j^\top \mathbf{x}_i)^2 \geq \frac{1}{4S} \; .$$

We sum the above inequality for every $i \in I_+$ to get:

$$\frac{|I_+|}{4S} \leq \sum_{j=1}^{n} \sum_{i=1}^{m} (\mathbf{w}_j^\top \mathbf{x}_i)^2$$

$$= \sum_{j=1}^{n} \sum_{i=1}^{m} \mathbf{w}_j^\top (\mathbf{x}_i \mathbf{x}_i^\top) \mathbf{w}_j$$

$$\leq \sum_{j=1}^{n} \|\mathbf{w}_j\|^2 \cdot \left\| \sum_{i=1}^{m} \mathbf{x}_i \mathbf{x}_i^\top \right\|$$

$$\leq \sum_{j=1}^{n} \|\mathbf{w}_j\|^2 \cdot C \leq S \cdot C$$

where in the second to last to last inequality we used the Property 2 for some constant $C > 0$, and in the last inequality we used that $\sum_{j=1}^{n} \|\mathbf{w}_j\|^2 \leq S$. Rearranging the above terms yields: $S \geq \sqrt{\frac{|I_+|}{12C}}$. $\qquad \square$

We now prove a lemma which constructs a specific solution that achieves a norm bound that depends on $|I_+|$.

**Lemma B.3.** *Assume $d \geq 50m^2 \log\left(\frac{3m^2}{\delta}\right)$ and $p \leq \frac{1}{4}$. There exists weights $\boldsymbol{\theta} = (\mathbf{w}_j, v_j, b_j)_{j=1}^{n}$ that attain a margin of at least $1$ on every sample and have $\sum_{j=1}^{n} \|\mathbf{w}_j\|^2 + v_j^2 + b_j^2 \leq 9\sqrt{|I_-|}$.*

*Proof.* Assume without loss of generality that $n$ is even (otherwise fix the last neuron to be 0). We consider the following weight assignment: For every $j \leq \frac{n}{2}$, consider $\mathbf{w}_j = -\sqrt{\frac{4}{n\sqrt{|I_-|}}} \sum_{i \in I_-} \mathbf{x}_i$, $v_j = 2\sqrt{\frac{\sqrt{|I_-|}}{n}}$ and $b_j = \sqrt{\frac{4}{n\sqrt{|I_-|}}}$. For every $j > \frac{n}{2}$, consider $\mathbf{w}_j = \sqrt{\frac{4}{n\sqrt{|I_-|}}} \sum_{i \in I_-} \mathbf{x}_i$, $v_j = -2\sqrt{\frac{\sqrt{|I_-|}}{n}}$ and $b_j = 0$.

We first show that this solution attains a margin of at least 1. For every $i \in I_+$ we have:

$$N_{\boldsymbol{\theta}}(\mathbf{x}_i) = \sum_{j=1}^{n} v_j \sigma(\mathbf{w}_j^\top \mathbf{x}_i + b_j)$$

$$= \sum_{j \leq n/2} 2\sqrt{\frac{\sqrt{|I_-|}}{n}} \sigma \left( \sqrt{\frac{4}{n\sqrt{|I_-|}}} - \sqrt{\frac{4}{n\sqrt{|I_-|}}} \sum_{r \in I_-} \mathbf{x}_r^\top \mathbf{x}_i \right)$$

$$- \sum_{j > n/2} 2\sqrt{\frac{\sqrt{|I_-|}}{n}} \sigma \left( \sqrt{\frac{4}{n\sqrt{|I_-|}}} \sum_{r \in I_-} \mathbf{x}_r^\top \mathbf{x}_i \right)$$

$$\geq \sum_{j \leq n/2} \frac{4}{n} \sigma \left( 1 - \frac{|I_-| \sqrt{2 \log \left( \frac{3m^2}{\delta} \right)}}{\sqrt{d}} \right) - \sum_{j > n/2} \frac{4}{n} \sigma \left( \frac{|I_-| \sqrt{2 \log \left( \frac{3m^2}{\delta} \right)}}{\sqrt{d}} \right)$$

$$\geq \frac{n}{2} \cdot \frac{4}{n} \cdot \left( 1 - \frac{1}{10} \right) - \frac{n}{2} \cdot \frac{4}{n} \cdot \frac{1}{10} \geq 1 \, ,$$

where we Property 1, that $p \leq \frac{1}{4}$ hence by Property 3 $|I_-| \leq \frac{m}{2}$ and our assumption on $m$ and $d$. For $i \in I_-$ we have:

$$N_{\boldsymbol{\theta}}(\mathbf{x}_i) = \sum_{j=1}^{n} v_j \sigma(\mathbf{w}_j^\top \mathbf{x}_i + b_j)$$

$$= \sum_{j \leq n/2} 2\sqrt{\frac{\sqrt{|I_-|}}{n}} \sigma \left( \sqrt{\frac{4}{n\sqrt{|I_-|}}} - \sqrt{\frac{4}{n\sqrt{|I_-|}}} \sum_{r \in I_-} \mathbf{x}_r^\top \mathbf{x}_i \right)$$

$$- \sum_{j > n/2} 2\sqrt{\frac{\sqrt{|I_-|}}{n}} \sigma \left( \sqrt{\frac{4}{n\sqrt{|I_-|}}} \sum_{r \in I_-} \mathbf{x}_r^\top \mathbf{x}_i \right)$$

$$= \sum_{j \leq n/2} \frac{4}{n} \sigma \left( - \sum_{r \in I_- \backslash \{i\}} \mathbf{x}_r^\top \mathbf{x}_i \right) - \sum_{j > n/2} \frac{4}{n} \sigma \left( 1 + \sum_{r \in I_- \backslash \{i\}} \mathbf{x}_r^\top \mathbf{x}_i \right)$$

$$\leq \sum_{j \leq n/2} \frac{4}{n} \sigma \left( - \frac{(|I_-| - 1) \sqrt{2 \log \left( \frac{3m^2}{\delta} \right)}}{\sqrt{d}} \right) - \sum_{j > n/2} \frac{4}{n} \sigma \left( 1 - \frac{(|I_-| - 1) \sqrt{2 \log \left( \frac{3m^2}{\delta} \right)}}{\sqrt{d}} \right)$$

$$\leq \frac{n}{2} \cdot \frac{4}{n} \cdot \frac{1}{10} - \frac{n}{2} \cdot \frac{4}{n} \cdot \left( 1 - \frac{1}{10} \right) \leq -1 \, ,$$

where again we used properties 1 and 3, that $p \leq \frac{1}{4}$ hence $|I_-| \leq \frac{m}{2}$ and our assumption on $m$ and $d$. This shows that this is indeed a feasible solution. We turn to calculate the norm of this solution. First we bound the following:

$$\left\| \sum_{i \in I_-} \mathbf{x}_i \right\|^2 = \sum_{i \in I_-} \| \mathbf{x}_i \|^2 + \sum_{i \neq j, \, i,j \in I_-} \mathbf{x}_i^\top \mathbf{x}_j$$

$$\leq |I_-| + |I_-|^2 \cdot \frac{\sqrt{2 \log \left( \frac{3m^2}{\delta} \right)}}{\sqrt{d}} \leq |I_-| \cdot \frac{11}{10} \, ,$$

where we used Property 1 and the assumption on $m$ and $d$. We now use the above calculation to bound the norm of our solution:

$$\sum_{j=1}^{n} \|\mathbf{w}_j\|^2 + v_j^2 + b_j^2 = n \cdot \frac{4}{n\sqrt{|I_-|}} \left\| \sum_{i \in I_-} \mathbf{x}_i \right\|^2 + \frac{2n\sqrt{|I_-|}}{n} + \frac{n}{2} \cdot \frac{4}{n\sqrt{|I_-|}}$$

$$\leq \frac{44\sqrt{|I_-|}}{10} + 2\sqrt{|I_-|} + \frac{2}{\sqrt{|I_-|}} \leq 9\sqrt{|I_-|}$$

$\square$

We are now ready to prove the main theorem of this subsection:

*Proof of Theorem 4.1.* Denote by $K := \sum_{j=1}^{n} \|\mathbf{w}_j\|^2 + v_j^2 + b_j^2$, and assume that $K \leq \frac{a}{\sqrt{p}} \|\boldsymbol{\theta}^*\|^2$, where $a$ will be chosen later and $\boldsymbol{\theta}^*$ is a solution to the max margin solution from Eq. (1). Assume on the way of contradiction that $\sum_{j=1}^{n} v_j \sigma(b_j) \leq \frac{1}{2}$, then by Lemma B.2 we know that: $K \geq C\sqrt{|I_+|} \geq C\sqrt{\left(1 - \frac{3p}{2}\right)m}$. On the other hand, by Lemma B.3 we know that $\|\boldsymbol{\theta}^*\| \leq 9\sqrt{|I_-|} \leq 9\sqrt{\frac{3p}{2}m}$. Combining this with the assumption we have on $K$ we get that:

$$C\sqrt{\left(1 - \frac{3p}{2}\right)m} \leq \frac{9a}{\sqrt{p}}\sqrt{\frac{3pm}{2}} .$$

Picking $a$ to be some constant with $a < \frac{C\sqrt{2}}{18\sqrt{3}}$ contradicts the above inequality. Thus, there exists a constant $c_4 := \frac{a}{2}$ such that if $K \leq \frac{c_4}{\sqrt{p}} S$ then $\sum_{j=1}^{n} v_j \sigma(b_j) > \frac{1}{2}$.

Suppose we sample $\mathbf{x} \sim \mathcal{N}\left(0, \frac{1}{d}I\right)$, then we have:

$$N_{\boldsymbol{\theta}}(\mathbf{x}) = \sum_{j=1}^{n} v_j \sigma(\mathbf{w}_j^\top \mathbf{x} + b_j)$$

$$= \sum_{j=1}^{n} v_j \sigma(b_j) - \left( \sum_{j=1}^{n} v_j \sigma(b_j) - \sum_{j=1}^{n} v_j \sigma(\mathbf{w}_j^\top \mathbf{x} + b_j) \right)$$

$$\geq \frac{1}{2} - \left( \sum_{j=1}^{n} v_j \sigma(b_j) - \sum_{j=1}^{n} v_j \sigma(\mathbf{w}_j^\top \mathbf{x} + b_j) \right)$$

$$\geq \frac{1}{2} - \sum_{j=1}^{n} |v_j| \cdot |\mathbf{w}_j^\top \mathbf{x}|$$

$$\geq \frac{1}{2} - \sqrt{\sum_{j=1}^{n} v_j^2} \sqrt{\sum_{j=1}^{n} (\mathbf{w}_j^\top \mathbf{x})^2} . \tag{17}$$

We will now bound the terms of the above equation. Note that $\sum_{j=1}^{n} v_j^2 \leq K \leq 9\sqrt{m}$. For the second term, we denote by $\bar{\mathbf{w}}_j := \frac{\mathbf{w}_j}{\|\mathbf{w}_j\|}$, and write:

$$\sum_{j=1}^{n} (\mathbf{w}_j^\top \mathbf{x})^2 = \sum_{j=1}^{n} \|\mathbf{w}_j\|^2 (\bar{\mathbf{w}}_j^\top \mathbf{x})^2$$

$$\leq \max_{j \in [n]} (\bar{\mathbf{w}}_j^\top \mathbf{x})^2 \cdot \sum_{j=1}^{n} \|\mathbf{w}_j\|^2 .$$

Again, we have that $\sum_{j=1}^{n} \|\mathbf{w}_j\|^2 \leq K \leq 9\sqrt{m}$. By using the stationarity KKT condition from Eq. (2), we get that $\bar{\mathbf{w}}_j \in \text{span}\{\mathbf{x}_1, \ldots, \mathbf{x}_m\}$, thus we can write $\bar{\mathbf{w}}_j = \sum_{i=1}^{m} \alpha_{i,j} \mathbf{x}_i = X\boldsymbol{\alpha}_j$ where

$X$ is a matrix with rows equal to $\mathbf{x}_i$, and $(\boldsymbol{\alpha}_j)_i = \alpha_{i,j}$. We will bound $a := \arg\max_i |\alpha_{i,j}|$:

$$
\begin{aligned}
1 = \|\bar{\mathbf{w}}_j\|^2 &= \left\| \sum_{i=1}^m \alpha_{i,j} \mathbf{x}_i \right\|^2 \\
&= \|X\boldsymbol{\alpha}_j\|^2 = \boldsymbol{\alpha}_j^\top X^\top X \boldsymbol{\alpha}_j \\
&= \boldsymbol{\alpha}_j^\top I \boldsymbol{\alpha}_j + \boldsymbol{\alpha}_j^\top \left( X^\top X - I \right) \boldsymbol{\alpha}_j \\
&\geq \|\boldsymbol{\alpha}_j\|^2 - \|\boldsymbol{\alpha}_j\|^2 \|X^\top X - I\| \\
&= \|\boldsymbol{\alpha}_j\|^2 - \|\boldsymbol{\alpha}_j\|^2 \|XX^\top - I\| ,
\end{aligned}
$$

where the last equality is true by using the SVD decomposition of $X$. Namely, write $X = USV^\top$, then $\|X^\top X - I\| = \|VS^2V^\top - I\| = \|S^2 - I\| = \|U^\top S^2 U - I\| = \|XX^\top - I\|$. Note that in Lemma B.1, Eq. (16) we have shown that $\|XX^\top - I\| \leq c'$ for some constant $c'$. Note that we can choose $m$ large enough such that $c' \leq \frac{1}{2}$. In total, this shows that $\|\boldsymbol{\alpha}_j\|^2 \leq c''$ for some constant $c''$.

We now use Lemma D.1 and the union bound to get that with probability $> 1 - \epsilon$ we have for every $i \in I$ that $|\mathbf{x}^\top \mathbf{x}_i| \leq \sqrt{\frac{2\log\left(\frac{m}{\epsilon}\right)}{d}}$. We condition on this event from now on. Applying both bounds we get:

$$
\begin{aligned}
\max_{j \in [n]} (\bar{\mathbf{w}}_j^\top \mathbf{x}) &\leq \max_{j \in [n]} \left( \sum_{i=1}^m \alpha_{i,j} |\mathbf{x}_i^\top \mathbf{x}| \right) \\
&\leq \sqrt{\frac{2\log\left(\frac{m}{\epsilon}\right)}{d}} \max_{j \in [n]} \|\boldsymbol{\alpha}_j\| \\
&\leq c'' \sqrt{\frac{2\log\left(\frac{m}{\epsilon}\right)}{d}} ,
\end{aligned}
$$

Plugging in the bounds above to Eq. (17) we get:

$$
\begin{aligned}
N_{\boldsymbol{\theta}}(\mathbf{x}) &\geq \frac{1}{2} - \sqrt{\sum_{j=1}^n v_j^2} \sqrt{\sum_{j=1}^n (\mathbf{w}_j^\top \mathbf{x})^2} \\
&\geq \frac{1}{2} - 9\sqrt{m} \cdot 9\sqrt{m} \cdot c'' \sqrt{\frac{2\log\left(\frac{m}{\epsilon}\right)}{d}} \\
&\geq \frac{1}{2} - \frac{81 m c'' \sqrt{2\log\left(\frac{m}{\epsilon}\right)}}{\sqrt{d}} .
\end{aligned}
$$

By choosing a constant $c_3 > 0$ large enough such that if $d \geq c_3 m^2 \sqrt{\log\left(\frac{m}{\epsilon}\right)}$ then , then $\frac{81 m c'' \sqrt{2\log\left(\frac{m}{\epsilon}\right)}}{\sqrt{d}} \leq \frac{1}{4}$ we get that $N_{\boldsymbol{\theta}}(\mathbf{x}) > 0$. $\qquad\square$

### B.2 Proof of Theorem 4.3

We sample a dataset $\mathbf{x}_1, \ldots, \mathbf{x}_m \sim \mathrm{Unif}(\mathbb{S}^{d-1})$, and labels $y_1, \ldots, y_m \sim \mathcal{D}_y$, for $p \leq c_1$ for some universal constant $c_1$. We first prove that the following properties holds with probability $> 1 - \delta$:

1. For every $i, j \in [m]$, $|\langle \mathbf{x}_i, \mathbf{x}_j \rangle| \leq \sqrt{\frac{2\log\left(\frac{2m^2}{\delta}\right)}{d}}$
2. $|I_-| \leq \frac{c_1 m}{n^2}$.

**Lemma B.4.** *Let $\delta > 0$, assume that we sample $\mathbf{x}_1, \ldots \mathbf{x}_m \sim \mathrm{Unif}(\mathbb{S}^{d-1})$, and $y_1, \ldots, y_m \sim \mathcal{D}_y$ for $p \leq \frac{c}{n^2}$, and that $m > c' \log\left(\frac{2}{\delta}\right)$, for some universal constant $c, c' > 0$. Then, with probability $> 1 - \delta$ properties 1 and 2 holds.*

The proof is the same as in Lemma B.1 so we will not repeat it for conciseness. The only different is that here we only need $|I_-|$ to be smaller than $\frac{c_1 m}{n^2}$ which is independent of $p$. Hence, for $p \leq \frac{c}{n^2}$ we get this concentration where $m$ depends only on the probability. From now on we condition on the event that properties 1 and 2 hold, and our bounds will depend on the probability that this event happens.

In this section we consider a networks of the form:

$$N_{\boldsymbol{\theta}}(\mathbf{x}) = \sum_{j=1}^{n/2} \sigma(\mathbf{w}_j^\top \mathbf{x} + b_j) - \sum_{j=n/2+1}^{n} \sigma(\mathbf{w}_j^\top \mathbf{x} + b_j)$$

That is, the output weights are all fixed to be $\pm 1$, equally divided between the neurons. We will use the stationarity condition (Eq. (2)) freely throughout the proof. For our network it means that we can write for every $j \in [n]$:

$$\mathbf{w}_j = v_j \sum_{i \in [m]} \lambda_i \sigma'_{i,j} y_i \mathbf{x}_i$$

$$b_j = v_j \sum_{i \in [m]} \lambda_i \sigma'_{i,j} y_i \,,$$

where $\sigma'_{i,j} = \mathbb{1}(\mathbf{w}_j^\top \mathbf{x}_i + b_j >)$, $v_j = 1$ for $j \in \{1, \ldots, n/2\}$ and $v_j = -1$ for $j \in \{n/2+1, \ldots, n\}$.

The next lemma shows that all the biases are positive. Note that this lemma relies only on that the dimension is large enough, and that Property 1 of the data holds:

**Lemma B.5.** *Assume that $d \geq 8m^4 \log\left(\frac{2m^2}{\delta}\right)$. Then for every $j \in [n]$ we have $b_j \geq 0$.*

*Proof.* Assume on the way of contradiction that $b_j < 0$ for some $j \in [n]$. We assume without loss of generality that $j \in \{1, \ldots, n/2\}$, the proof for $j \in \{n/2 + 1, \ldots, n\}$ is done similarly. Denote by $I'_+ = \{i \in I_+ := \sigma_{i,j} = 1\}$ and $I'_- = \{i \in I_- := \sigma_{i,j} = 1\}$. We can write:

$$b_j = \sum_{i \in [m]} \lambda_i \sigma'_{i,j} y_i = \sum_{i \in I'_+} \lambda_i - \sum_{i \in I'_-} \lambda_i \,.$$

Note that $\lambda_i \geq 0$ for every $i$, hence $I'_-$ is non empty, otherwise $b_j \geq 0$. Take $\lambda_r = \arg\max_{i \in I'_-} \lambda_i$, we have:

$$0 < w_j^\top \mathbf{x}_r + b_j = b_j + \sum_{i \in I'_+ \cup I'_-} \lambda_i y_i \mathbf{x}_r^\top \mathbf{x}_i$$

$$< -\lambda_r + \sum_{i \in I'_+ \cup I'_- \setminus \{r\}} \lambda_i y_i \mathbf{x}_i^\top \mathbf{x}_r \,,$$

where we used that $b_j < 0$ and $\|\mathbf{x}_r\| = 1$. Rearranging the terms and using Property 1:

$$\lambda_r < \sum_{i \in I'_+ \cup I'_- \setminus \{r\}} \lambda_i |y_i \mathbf{x}_i^\top \mathbf{x}_r|$$

$$\leq \sqrt{\frac{2 \log\left(\frac{2m^2}{\delta}\right)}{d}} \sum_{i \in I'_+ \cup I'_- \setminus \{r\}} \lambda_i$$

$$\leq \sqrt{\frac{2 \log\left(\frac{2m^2}{\delta}\right)}{d}} \sum_{i \in I'_+ \cup I'_-} \lambda_i \,.$$

Denote $\lambda_s := \arg\max_{i \in I'_+ \cup I'_-} \lambda_i$, then from the above we have shown that $\lambda_r \leq \lambda_s m \sqrt{\frac{2 \log\left(\frac{2m^2}{\delta}\right)}{d}}$, since $|I'_+ \cup I'_-| \leq m$. In particular, $s \in I'_+$, otherwise, $\lambda_r = \lambda_s$ which means that $\lambda_r < 0$ since

$m\sqrt{\frac{2\log\left(\frac{2m^2}{\delta}\right)}{d}} < 1$ which is a contradiction. We can now write:

$$b_j = \sum_{i \in I'_+} \lambda_i - \sum_{i \in I'_-} \lambda_i$$

$$\geq \sum_{i \in I'_+} \lambda_i - \sum_{i \in I'_-} \lambda_r$$

$$\geq \lambda_s - \lambda_s m^2 \sqrt{\frac{2\log\left(\frac{2m^2}{\delta}\right)}{d}}$$

$$= \lambda_s \left(1 - m^2 \sqrt{\frac{2\log\left(\frac{2m^2}{\delta}\right)}{d}}\right) .$$

By our assumption on $d$, we have that $m^2\sqrt{\frac{2\log\left(\frac{2m^2}{\delta}\right)}{d}} \leq \frac{1}{2}$, hence $b_j \geq \frac{\lambda_s}{2}$. But, by our assumption, $b_j < 0$ which is a contradiction since $\lambda_s \geq 0$. $\qquad\square$

The following lemma is a general property of the KKT conditions:

**Lemma B.6.** *Let $i \in I_+$ (resp. $i \in I_-$) with $\lambda_i > 0$. Then, there is a neuron $k$ with positive output weight (resp. negative output weight) s.t $\sigma'_{i,k} = 1$.*

*Proof.* We prove it for $i \in I_+$, the other case is similar. Assume otherwise, that is for every neurons with positive output $j$ we have $\sigma'_{i,j} = 0$, that is $\mathbf{w}_j^\top \mathbf{x}_i + b_j \leq 0$ by the definition of $\sigma'_{i,j}$. This means that $N_{\boldsymbol{\theta}}(\mathbf{x}_i) \leq 0$, since all the positive neurons are inactive on $\mathbf{x}_i$, which is a contradiction to $N_{\boldsymbol{\theta}}(\mathbf{x}_i) \geq 1$. $\qquad\square$

The next lemma shows that if the bias terms are smaller than $\frac{1}{4}$, then the $\lambda_i$'s for $i \in I_+$ cannot be too small.

**Lemma B.7.** *Assume that $d \geq 32m^4 n^2 \log\left(\frac{2m^2}{\delta}\right)$, and that $\sum_{j=1}^{n/2} b_j - \sum_{j=n/2+1}^{n} b_j \leq \frac{1}{4}$. Then, for every $i \in I_+$ we have $\lambda_i \geq \frac{1}{4n}$.*

*Proof.* Take $r \in I_+$, we have:

$$1 \leq N_{\boldsymbol{\theta}}(\mathbf{x}_r) = \sum_{j=1}^{n/2} \sigma(\mathbf{w}_j^\top \mathbf{x}_r + b_j) - \sum_{j=n/2+1}^{n} \sigma(\mathbf{w}_j^\top \mathbf{x}_r + b_j)$$

$$= \sum_{j=1}^{n/2} \sigma\left(\lambda_r \sigma'_{r,j} + \sum_{i \in I \setminus \{r\}} \lambda_i y_i \sigma'_{i,j} \mathbf{x}_i^\top \mathbf{x}_r + b_j\right)$$

$$- \sum_{j=n/2+1}^{n} \sigma\left(-\lambda_r \sigma'_{r,j} - \sum_{i \in I \setminus \{r\}} \lambda_i y_i \sigma'_{i,j} \mathbf{x}_i^\top \mathbf{x}_r + b_j\right)$$

$$\leq \sum_{j=1}^{n/2} \sigma\left(\lambda_r + \sum_{i \in I \setminus \{r\}} \lambda_i \sigma'_{i,j} |\mathbf{x}_i^\top \mathbf{x}_r| + b_j\right) - \sum_{j=n/2+1}^{n} \sigma\left(-\lambda_r - \sum_{i \in I \setminus \{r\}} \lambda_i \sigma'_{i,j} |\mathbf{x}_i^\top \mathbf{x}_r| + b_j\right)$$

$$\tag{18}$$

We will bound the two terms above. For the first term, note that all the terms inside the ReLU are positive, since by Lemma B.5 we have $b_j \geq 0$, and $\lambda_i \geq 0$ by Eq. (4) hence we can remove the ReLU

function. Using Property 1 we can bound:

$$\sum_{j=1}^{n/2} \sigma\left(\lambda_r + \sum_{i\in I\setminus\{r\}} \lambda_i\sigma'_{i,j}|\mathbf{x}_i^\top\mathbf{x}_r| + b_j\right) \le \frac{n}{2}\lambda_r + \sqrt{\frac{2\log\left(\frac{2m^2}{\delta}\right)}{d}}\sum_{j=1}^{n/2}\sum_{i\in I\setminus\{r\}}\lambda_i\sigma'_{i,j} + \sum_{j=1}^{n/2} b_j$$

$$\le \frac{n}{2}\lambda_r + \frac{n}{2}\cdot\sqrt{\frac{2\log\left(\frac{2m^2}{\delta}\right)}{d}}\sum_{i\in I\setminus\{r\}}\lambda_i + \sum_{j=1}^{n/2}b_j \ , \quad (19)$$

where we used that $\sigma'_{i,j} \le 1$. For the second term in Eq. (18) we use the fact that the ReLU function is 1-Lipschitz to get that:

$$-\sum_{j=n/2+1}^{n}\sigma\left(-\lambda_r - \sum_{i\in I\setminus\{r\}}\lambda_i\sigma'_{i,j}|\mathbf{x}_i^\top\mathbf{x}_r| + b_j\right)$$

$$= -\sum_{j=n/2+1}^{n}\sigma\left(-\lambda_r - \sum_{i\in I\setminus\{r\}}\lambda_i\sigma'_{i,j}|\mathbf{x}_i^\top\mathbf{x}_r| + b_j\right) + \sum_{j=n/2+1}^{n}\sigma(b_j) - \sum_{j=n/2+1}^{n}\sigma(b_j)$$

$$\le -\sum_{j=n/2+1}^{n} b_j + \sum_{j=n/2+1}^{n}\left|\lambda_r + \sum_{i\in I\setminus\{r\}}\lambda_i\sigma'_{i,j}|\mathbf{x}_i^\top\mathbf{x}_r|\right|$$

$$\le -\sum_{j=n/2+1}^{n} b_j + \frac{n}{2}\lambda_r + \frac{n}{2}\cdot\sqrt{\frac{2\log\left(\frac{2m^2}{\delta}\right)}{d}}\sum_{i\in I\setminus\{r\}}\lambda_i \ , \quad (20)$$

where we again used Lemma B.5 to get that $b_j \ge 0$, and Property 1. Combining Eq. (19) and Eq. (20) with Eq. (18) we get that:

$$1 \le \sum_{j=1}^{n/2} b_j - \sum_{j=n/2+1}^{n} b_j + n\lambda_r + n\sqrt{\frac{2\log\left(\frac{2m^2}{\delta}\right)}{d}}\sum_{i\in I\setminus\{r\}}\lambda_i \ .$$

Rearranging the terms above, and using our assumption on the biases we get that:

$$\frac{3}{4} \le n\lambda_r + n\sqrt{\frac{2\log\left(\frac{2m^2}{\delta}\right)}{d}}\sum_{i\in I\setminus\{r\}}\lambda_i \ .$$

If $\lambda_r \ge \frac{1}{4n}$ then we are done. Assume otherwise, then $n\sqrt{\frac{2\log\left(\frac{2m^2}{\delta}\right)}{d}}\sum_{i\in I\setminus\{r\}}\lambda_i \ge \frac{1}{2}$ and $\lambda_r < \frac{1}{4n}$. Denote $\lambda_s := \arg\max_{i\in[m]}\lambda_i$, then we have that $\sum_{i\in[m]\setminus\{r\}}\lambda_i \le m\lambda_s$, which by the above inequality means that $\lambda_s \ge \frac{\sqrt{d}}{2mn\sqrt{2\log\left(\frac{2m^2}{\delta}\right)}}$. We split into cases:

**Case I.** Assume $s \in I_+$. By Lemma B.6 there is $k \in \{1,\ldots,n/2\}$ with $\sigma'_{s,k} = 1$. For the $k$-th neuron we have:

$$\sigma(\mathbf{w}_k^\top\mathbf{x}_s + b_k) = \sigma\left(\sum_{i\in[m]} y_i\lambda_i\sigma'_{i,k}\mathbf{x}_i^\top\mathbf{x}_s + b_k\right)$$

$$\ge \sigma\left(\lambda_s - \sum_{i\in I\setminus\{s\}}\lambda_i|\mathbf{x}_i^\top\mathbf{x}_s| + b_k\right)$$

$$\ge \sigma\left(\lambda_s - m\lambda_s\sqrt{\frac{2\log\left(\frac{2m^2}{\delta}\right)}{d}} + b_k\right) \ , \quad (21)$$

and note that since $\lambda_s \geq 0$, $m\sqrt{\frac{2\log\left(\frac{2m^2}{\delta}\right)}{d}} < 1$ and $b_j \geq 0$ this neuron is active. For every other neuron $j$ with a positive output weight we have:

$$\sigma(\mathbf{w}_j^\top \mathbf{x}_s + b_j) - \sigma(b_j) = \sigma\left(\sum_{i\in[m]} y_i \lambda_i \sigma'_{i,j} \mathbf{x}_i^\top \mathbf{x}_s + b_j\right) - \sigma(b_j)$$

$$\geq \sigma\left(\sum_{i\in[m]} \lambda_i |\mathbf{x}_i^\top \mathbf{x}_s| + b_j\right) - \sigma(b_j)$$

$$\geq \sigma\left(-m\lambda_s \sqrt{\frac{2\log\left(\frac{2m^2}{\delta}\right)}{d}} + b_j\right) - \sigma(b_j)$$

$$\geq -m\lambda_s \sqrt{\frac{2\log\left(\frac{2m^2}{\delta}\right)}{d}} , \tag{22}$$

where we used that $\sigma$ is 1-Lipschitz and that $\lambda_s$ is the largest among the $\lambda_i$'s. For a neuron with a negative output weight, we can write:

$$\sigma(\mathbf{w}_j^\top \mathbf{x}_s + b_j) = \sigma\left(-\sum_{i\in[m]} y_i \lambda_i \sigma'_{i,j} \mathbf{x}_i^\top \mathbf{x}_s + b_j\right)$$

$$\leq \sigma\left(-\sum_{i\in I\setminus\{s\}} \lambda_i |\mathbf{x}_i^\top \mathbf{x}_s| + b_j\right) \leq b_j + m\lambda_s \sqrt{\frac{2\log\left(\frac{2m^2}{\delta}\right)}{d}}$$

In total, combining the above bound with Eq. (21) and Eq. (22) we have that:

$$1 = N_{\boldsymbol{\theta}}(\mathbf{x}_s) = \sum_{j=1}^{n/2} \sigma(\mathbf{w}_j^\top \mathbf{x}_s + b_j) - \sum_{j=n/2+1}^{n} \sigma(\mathbf{w}_j^\top \mathbf{x}_s + b_j)$$

$$= \sum_{j=1}^{n/2} \sigma(\mathbf{w}_j^\top \mathbf{x}_s + b_j) - \sum_{j=n/2+1}^{n} \sigma(\mathbf{w}_j^\top \mathbf{x}_s + b_j) + \sum_{j=1}^{n/2} \sigma(b_j) - \sum_{j=1}^{n/2} \sigma(b_j)$$

$$\geq -\lambda_s mn \sqrt{\frac{2\log\left(\frac{2m^2}{\delta}\right)}{d}} + \lambda_s + \sum_{j=1}^{n/2} b_j - \sum_{j=n/2+1}^{n} b_j$$

By our assumption on $d$ we have that $mn\sqrt{\frac{2\log\left(\frac{2m^2}{\delta}\right)}{d}} \leq \frac{1}{2}$. Hence, rearranging the terms above, we get that:

$$\sum_{j=1}^{n/2} b_j - \sum_{j=n/2+1}^{n} b_j \leq 1 - \frac{\lambda_s}{2} .$$

But then, looking at the output of the network on $\lambda_r$ (recall that by our assumption $\lambda_r \leq \frac{1}{4n}$) we get:

$$
\begin{aligned}
1 \leq N_{\boldsymbol{\theta}}(\mathbf{x}_r) &= \sum_{j=1}^{n/2} \sigma(\mathbf{w}_j^\top \mathbf{x}_r + b_j) - \sum_{j=n/2+1}^{n} \sigma(\mathbf{w}_j^\top \mathbf{x}_r + b_j) \\
&= \sum_{j=1}^{n/2} \sigma\left(\sum_{i\in[m]} y_i \lambda_i \sigma'_{i,j} \mathbf{x}_i^\top \mathbf{x}_r + b_j\right) - \sum_{j=n/2+1}^{n} \sigma\left(-\sum_{i\in[m]} y_i \lambda_i \sigma'_{i,j} \mathbf{x}_i^\top \mathbf{x}_r + b_j\right) + \\
&\quad + \sum_{j=n/2+1}^{n} \sigma(b_j) - \sum_{j=n/2+1}^{n} \sigma(b_j) \\
&\leq \sum_{j=1}^{n/2} \sigma\left(\lambda_r + \sum_{i\in I\setminus\{r\}} \lambda_i |\mathbf{x}_i^\top \mathbf{x}_r| + b_j\right) - \sum_{j=n/2+1}^{n} \sigma\left(-\sum_{i\in I\setminus\{r\}} \lambda_i |\mathbf{x}_i^\top \mathbf{x}_r| + b_j\right) + \\
&\quad + \sum_{j=n/2+1}^{n} \sigma(b_j) - \sum_{j=n/2+1}^{n} \sigma(b_j) \\
&\leq \frac{n\lambda_r}{2} + \lambda_s mn\sqrt{\frac{2\log\left(\frac{2m^2}{\delta}\right)}{d}} + \sum_{j=1}^{n/2} b_j - \sum_{j=n/2+1}^{n} b_j \\
&\leq \frac{1}{8} + \lambda_s mn\sqrt{\frac{2\log\left(\frac{2m^2}{\delta}\right)}{d}} + 1 - \frac{\lambda_s}{2} \\
&\leq \frac{9}{8} + \lambda_s \left(mn\sqrt{\frac{2\log\left(\frac{2m^2}{\delta}\right)}{d}} - \frac{1}{2}\right).
\end{aligned}
\tag{23}
$$

By our assumption $mn\sqrt{\frac{2\log\left(\frac{2m^2}{\delta}\right)}{d}} \leq \frac{1}{4}$, hence rearranging the terms above we get that $\lambda_s \leq \frac{1}{2}$ which is a contradiction to $\lambda_s \geq \frac{\sqrt{d}}{2mn\sqrt{2\log\left(\frac{2m^2}{\delta}\right)}} > 1$.

**Case II.** Assume $s \in I_-$. By Lemma B.6 there is $k \in \{n/2 + 1, \ldots, n\}$ with $\sigma'_{s,k} = 1$. Note that since $\lambda_s \neq 0$ we have that $N_{\boldsymbol{\theta}}(\mathbf{x}_s) = -1$ (i.e this sample is on the margin). We have that:

$$
\begin{aligned}
-1 = N_{\boldsymbol{\theta}}(\mathbf{x}_s) &= \sum_{j=1}^{n/2} \sigma(\mathbf{w}_j^\top \mathbf{x}_s + b_j) - \sum_{j=n/2+1}^{n} \sigma(\mathbf{w}_j^\top \mathbf{x}_s + b_j) \\
&= \sum_{j=1}^{n/2} \sigma\left(\sum_{i\in[m]} y_i \lambda_i \sigma'_{i,j} \mathbf{x}_i^\top \mathbf{x}_s + b_j\right) - \sum_{j=n/2+1}^{n} \sigma\left(-\sum_{i\in[m]} y_i \lambda_i \sigma'_{i,j} \mathbf{x}_i^\top \mathbf{x}_s + b_j\right) + \\
&\quad + \sum_{j=n/2+1}^{n} \sigma(b_j) - \sum_{j=n/2+1}^{n} \sigma(b_j) \\
&\leq \sum_{j=1}^{n/2} \sigma\left(\sum_{i\in I\setminus\{s\}} \lambda_i |\mathbf{x}_i^\top \mathbf{x}_s| + b_j\right) - \sum_{j=n/2+1}^{n} \sigma\left(\lambda_s - \sum_{i\in I\setminus\{s\}} \lambda_i |\mathbf{x}_i^\top \mathbf{x}_s| + b_j\right) + \\
&\quad + \sum_{j=n/2+1}^{n} \sigma(b_j) - \sum_{j=n/2+1}^{n} \sigma(b_j) \\
&\leq \sum_{j=1}^{n/2} b_j - \sum_{j=n/2+1}^{n} b_j + \lambda_s mn\sqrt{\frac{2\log\left(\frac{2m^2}{\delta}\right)}{d}} - \lambda_s \\
&\leq \frac{1}{4} + \lambda_s \left(mn\sqrt{\frac{2\log\left(\frac{2m^2}{\delta}\right)}{d}} - 1\right), \quad\quad (24)
\end{aligned}
$$

where we used that $mn\sqrt{\frac{2\log\left(\frac{2m^2}{\delta}\right)}{d}} \leq \frac{1}{2}$ and $b_j \geq 0$ by Lemma B.5, hence in the second to last inequality the term inside the ReLU is positive. In addition, we used that the ReLU function is 1-Lipschitz, and that $\lambda_s$ is the largest amont the $\lambda_i$'s. Since $mn\sqrt{\frac{2\log\left(\frac{2m^2}{\delta}\right)}{d}} \leq \frac{1}{2}$, rearranging the terms above give us that $\lambda_s \leq \frac{10}{4}$, which is a contradiction to that $\lambda_s \geq \frac{\sqrt{d}}{2mn\sqrt{2\log\left(\frac{2m^2}{\delta}\right)}} \geq 3$.

To conclude, both cases are not possible, hence $\lambda_r \geq \frac{1}{4n}$, which is true for every $r \in I_+$.

$\square$

**Lemma B.8.** *Assume that $d \geq 16m^4 n^4 \log\left(\frac{2m^2}{\delta}\right)$, and that $\sum_{j=1}^{n/2} b_j - \sum_{j=n/2+1}^{n} b_j \leq \frac{1}{4}$. Then, for every $i \in I_-$ we have $\lambda_i \leq \frac{10}{4}$.*

*Proof.* Take $\lambda_s := \arg\max_{i\in[m]} \lambda_i$. We split into two cases:

**Case I.** Assume $s \in I_-$. Note that $\lambda_s > 0$, otherwise $\lambda_i = 0$ for every $i$, which means that $N_{\boldsymbol{\theta}}(\mathbf{x})$ is the zero function. Hence, $\mathbf{x}_s$ lies on the margin, and also by Lemma B.6 there is a neuron $j$ with a negative output weight such that $\sigma'_{s,j} = 1$. By the same calculation as in Eq. (24) we have::

$$
\begin{aligned}
-1 = N_{\boldsymbol{\theta}}(\mathbf{x}_s) &= \sum_{j=1}^{n/2} \sigma(\mathbf{w}_j^\top \mathbf{x}_s + b_j) - \sum_{j=n/2+1}^{n} \sigma(\mathbf{w}_j^\top \mathbf{x}_s + b_j) \\
&\leq \frac{1}{4} + \lambda_s \left(mn\sqrt{\frac{2\log\left(\frac{2m^2}{\delta}\right)}{d}} - 1\right)
\end{aligned}
$$

Using our assumption that $mn\sqrt{\frac{2\log\left(\frac{2m^2}{\delta}\right)}{d}} \leq \frac{1}{2}$ and rearranging the above terms we get: $\lambda_s \leq \frac{10}{4}$ which finishes the proof.

**Case II.** Assume $s \in I_+$. Denote by $\lambda_r := S \arg\max_{i \in I_-} \lambda_i$. By Lemma B.6 there is at least one neuron $k$ with a positive output weight such that $\sigma'_{s,k} = 1$, for this neuron we can bound:

$$b_k = \sum_{i \in [m]} y_i \lambda_i \sigma'_{i,k} \geq \lambda_s - m\lambda_r .$$

Every neuron $j$ with a negative output weight can be bounded similarly by:

$$b_j = -\sum_{i \in [m]} y_i \lambda_i \sigma'_{i,j} \leq m\lambda_r .$$

Combining the above bounds, and using that $b_j \geq 0$ by Lemma B.5 we can bound:

$$\frac{1}{4} \geq \sum_{j=1}^{n/2} b_j - \sum_{j=n/2+1}^{n} b_j$$

$$\geq \lambda_s - m\lambda_r - \sum_{j=n/2+1}^{n} m\lambda_r$$

$$\geq \lambda_s - mn\lambda_r .$$

Rearranging the terms we get that: $\lambda_s \leq mn\lambda_r + \frac{1}{4}$. If $\lambda_r \leq 1$ we are finished, otherwise $\lambda_s \leq 2mn\lambda_r$ and also $\mathbf{x}_r$ lies on the margin since $\lambda_r > 0$, hence. By doing a similar calculation to Case I we can write:

$$-1 = N_{\boldsymbol{\theta}}(\mathbf{x}_r) = \sum_{j=1}^{n/2} \sigma(\mathbf{w}_j^\top \mathbf{x}_r + b_j) - \sum_{j=n/2+1}^{n} \sigma(\mathbf{w}_j^\top \mathbf{x}_r + b_j)$$

$$= \sum_{j=1}^{n/2} \sigma\left(\sum_{i \in [m]} y_i \lambda_i \sigma'_{i,j} \mathbf{x}_i^\top \mathbf{x}_r + b_j\right) - \sum_{j=n/2+1}^{n} \sigma\left(-\sum_{i \in [m]} y_i \lambda_i \sigma'_{i,j} \mathbf{x}_i^\top \mathbf{x}_r + b_j\right) +$$

$$+ \sum_{j=n/2+1}^{n} \sigma(b_j) - \sum_{j=n/2+1}^{n} \sigma(b_j)$$

$$\leq \sum_{j=1}^{n/2} \sigma\left(\sum_{i \in I \setminus \{r\}} \lambda_i |\mathbf{x}_i^\top \mathbf{x}_r| + b_j\right) - \sum_{j=n/2+1}^{n} \sigma\left(\lambda_r - \sum_{i \in I \setminus \{r\}} \lambda_i |\mathbf{x}_i^\top \mathbf{x}_r| + b_j\right) +$$

$$+ \sum_{j=n/2+1}^{n} \sigma(b_j) - \sum_{j=n/2+1}^{n} \sigma(b_j)$$

$$\leq -\lambda_r + \sum_{j=1}^{n/2} b_j - \sum_{j=n/2+1}^{n} b_j + \lambda_s mn\sqrt{\frac{2\log\left(\frac{2m^2}{\delta}\right)}{d}}$$

$$\leq -\lambda_r + \frac{1}{4} + 2\lambda_r m^2 n^2 \sqrt{\frac{2\log\left(\frac{2m^2}{\delta}\right)}{d}} .$$

By the assumption on $d$ we have $2m^2 n^2 \sqrt{\frac{2\log\left(\frac{2m^2}{\delta}\right)}{d}} \leq \frac{1}{2}$. Thus, rearranging the terms above and using these bounds we get that $\lambda_r \leq \frac{10}{4}$.

$\square$

**Lemma B.9.** *Assume that $p \leq \frac{c_1}{n^2}$, $m \geq c_2 n \log\left(\frac{1}{\delta}\right)$ and $d \geq 32m^4 n^4 \log\left(\frac{2m^2}{\delta}\right)$. Then, $\sum_{j=1}^{n/2} b_j - \sum_{j=n/2+1}^{n} b_j > \frac{1}{4}$.*

*Proof.* Assume on the way of contradiction that $\sum_{j=1}^{n/2} b_j - \sum_{j=n/2+1}^{n} b_j \leq \frac{1}{4}$. By Lemma B.7 we have that $\lambda_i \geq \frac{1}{4n}$ for every $i \in I_+$, and by Lemma B.8 we have that $\lambda_i \leq \frac{10}{4}$ for every $i \in I_-$. By

Lemma B.6, for every $i \in I_+$ there is some $j \in \{1, \ldots, n/2\}$ with $\sigma'_{i,j} = 1$. This means that:

$$\sum_{j=1}^{n/2} b_j = \sum_{j=1}^{n/2} \sum_{i \in [m]} y_i \lambda_i \sigma'_{i,j}$$

$$\geq \frac{|I_+|}{4n} - \sum_{j=1}^{n/2} \sum_{i \in I_-} \lambda_i \geq \frac{|I_+|}{4n} - \frac{10n|I_-|}{8} \ .$$

In a similar manner, we can bound:

$$\sum_{j=n/2+1}^{n} b_j = \sum_{j=n/2+1}^{n} \sum_{i \in [m]} y_i \lambda_i \sigma'_{i,j}$$

$$\geq \sum_{j=n/2+1}^{n} \sum_{i \in I_-} \lambda_i \geq -\frac{10n|I_-|}{8} \ .$$

Combining the two bounds above we get that:

$$\sum_{j=1}^{n/2} b_j - \sum_{j=n/2+1}^{n} b_j \geq \frac{|I_+|}{4n} - \frac{10n|I_-|}{4} \ .$$

But, by our assumptions we have that $m \geq 2n$ (for a large enough constant), and by Lemma B.4 we have $|I_-| \leq \frac{cm}{n^2}$ and $|I_+| \geq \left(1 - \frac{c}{n^2}\right) m$ for some universal constant $c > 0$ which we will later choose. Plugging these bounds to the displayed equation above, we get that:

$$\sum_{j=1}^{n/2} b_j - \sum_{j=n/2+1}^{n} b_j \geq \frac{1}{2} \cdot \left(1 - \frac{c}{n^2}\right) - \frac{10c}{2} \geq \frac{1}{2} - \frac{11c}{2} \ ,$$

where in the last inequality we used that $n \geq 1$. Taking $c$ to be a small enough constant (i.e. $c < \frac{1}{22}$), we get that $\sum_{j=1}^{n/2} b_j - \sum_{j=n/2+1}^{n} b_j > \frac{1}{4}$, which is a contradiction. $\qquad \square$

We are now ready to prove the main theorem of this section:

*Proof of Theorem 4.3.* By Lemma B.9 we have that $\sum_{j=1}^{n/2} b_j - \sum_{j=n/2+1}^{n} b_j > \frac{1}{4}$. Denote by $\lambda_s := \arg\max_{i \in [m]} \lambda_i$, we would first like to show that $\lambda_s \leq 3mn$. We split into two cases:

**Case I.** Assume $s \in I_+$. Note that $\mathbf{x}_s$ lies on the margin, otherwise $\lambda_i = 0$ for every $i \in I$, which means that $N_{\boldsymbol{\theta}}$ is the zero predictor, this contradicts the assumption that $N_{\boldsymbol{\theta}}$ classifies the data correctly. Using a similar analysis to Case I in Lemma B.7 we have:

$$1 = N_{\boldsymbol{\theta}}(\mathbf{x}_s) = \sum_{j=1}^{n/2} \sigma(\mathbf{w}_j^\top \mathbf{x}_s + b_j) - \sum_{j=n/2+1}^{n} \sigma(\mathbf{w}_j^\top \mathbf{x}_s + b_j)$$

$$= \sum_{j=1}^{n/2} \sigma\left(\sum_{i \in [m]} y_i \lambda_i \sigma'_{i,j} \mathbf{x}_i^\top \mathbf{x}_s + b_j\right) - \sum_{j=n/2+1}^{n} \sigma\left(-\sum_{i \in [m]} y_i \lambda_i \sigma'_{i,j} \mathbf{x}_i^\top \mathbf{x}_s + b_j\right) + \sum_{j=1}^{n/2} \sigma(b_j) - \sum_{j=1}^{n/2} \sigma(b_j)$$

$$\geq \lambda_s + \sum_{j=1}^{n/2} \sigma\left(-\sum_{i \in I \setminus \{s\}} \lambda_i |\mathbf{x}_i^\top \mathbf{x}_s| + b_j\right) - \sum_{j=n/2+1}^{n} \sigma\left(\sum_{i \in I \setminus \{s\}} \lambda_i |\mathbf{x}_i^\top \mathbf{x}_s| + b_j\right) + \sum_{j=1}^{n/2} \sigma(b_j) - \sum_{j=1}^{n/2} \sigma(b_j)$$

$$\geq \sum_{j=1}^{n/2} b_j - \sum_{j=n/2+1}^{n} b_j + \lambda_s - \lambda_s mn \sqrt{\frac{2 \log\left(\frac{2m^2}{\delta}\right)}{d}}$$

$$\geq \frac{1}{4} + \lambda_s \left(1 - mn \sqrt{\frac{2 \log\left(\frac{2m^2}{\delta}\right)}{d}}\right) \ ,$$

where we used Lemma B.6 to show that there is at least one $k \in \{1, \ldots, n/2\}$ with $\sigma'_{s,k} = 1$. Using that $mn\sqrt{\frac{2\log\left(\frac{2m^2}{\delta}\right)}{d}} \leq \frac{1}{2}$ for a large enough constant $c_3$ and rearranging the terms above we have that $\lambda_s \leq \frac{6}{4} \leq 3mn$ since $m, n \geq 1$.

**Case II.** Assume $s \in I_-$. Take $\lambda_r := \arg\max_{i \in I_+} \lambda_i$. By Lemma B.5 there is at least one neuron $k \in \{n/2+1, \ldots, n\}$ with $\sigma'_{s,k} = 1$. We have that:

$$
\begin{aligned}
\frac{1}{4} &\leq \sum_{j=1}^{n/2} b_j - \sum_{j=n/2+1}^{n} b_j \\
&\leq \sum_{j=1}^{n/2} \sum_{i \in [m]} \lambda_i y_i \sigma'_{i,j} - \sum_{j=n/2+1}^{n} \sum_{i \in [m]} \lambda_i y_i \sigma'_{i,j} \leq mn\lambda_r - \lambda_s .
\end{aligned}
$$

Rearranging the terms we get $\lambda_s \leq mn\lambda_r - \frac{1}{4} \leq mn\lambda_r$. Note that $\mathbf{x}_r$ lies on the margin, otherwise $\lambda_s = 0$, which similarly to Case $I$ is a contradiction. Again, using a similar analysis to Case I we get:

$$
\begin{aligned}
1 = N_{\boldsymbol{\theta}}(\mathbf{x}_r) &= \sum_{j=1}^{n/2} \sigma(\mathbf{w}_j^\top \mathbf{x}_r + b_j) - \sum_{j=n/2+1}^{n} \sigma(\mathbf{w}_j^\top \mathbf{x}_r + b_j) \\
&= \sum_{j=1}^{n/2} \sigma\left(\sum_{i \in [m]} y_i \lambda_i \sigma'_{i,j} \mathbf{x}_i^\top \mathbf{x}_s + b_j\right) - \sum_{j=n/2+1}^{n} \sigma\left(-\sum_{i \in [m]} y_i \lambda_i \sigma'_{i,j} \mathbf{x}_i^\top \mathbf{x}_s + b_j\right) + \sum_{j=1}^{n/2} \sigma(b_j) - \sum_{j=1}^{n/2} \sigma(b_j) \\
&\geq \lambda_r + \sum_{j=1}^{n/2} \sigma\left(-\sum_{i \in I \setminus \{s\}} \lambda_i |\mathbf{x}_i^\top \mathbf{x}_s| + b_j\right) - \sum_{j=n/2+1}^{n} \sigma\left(\sum_{i \in I \setminus \{s\}} \lambda_i |\mathbf{x}_i^\top \mathbf{x}_s| + b_j\right) + \sum_{j=1}^{n/2} \sigma(b_j) - \sum_{j=1}^{n/2} \sigma(b_j) \\
&\geq \sum_{j=1}^{n/2} b_j - \sum_{j=n/2+1}^{n} b_j + \lambda_r - \lambda_s mn\sqrt{\frac{2\log\left(\frac{2m^2}{\delta}\right)}{d}} \\
&\geq \frac{1}{4} + \lambda_r - \lambda_r m^2 n^2 \sqrt{\frac{2\log\left(\frac{2m^2}{\delta}\right)}{d}} ,
\end{aligned}
$$

By our assumption, $\frac{m^2 n^2 \log(d) 2\sqrt{2}}{\sqrt{d}} \leq \frac{1}{2}$ for a large enough constant $c_3$. Hence, rearranging the terms we get $\lambda_r \leq \frac{6}{4}$. This means that $\lambda_s \leq 2mn\lambda_r \leq 3mn$.

We now turn to calculate the output of $N_{\boldsymbol{\theta}}$ on the sample $\mathbf{x}$. Suppose we sample $\mathbf{x} \sim \text{Unif}(\mathbb{S}_{d-1})$, by Lemma D.1 we have with probability $> 1 - \epsilon$ that $|\mathbf{x}_i^\top \mathbf{x}| \leq \sqrt{\frac{2\log\left(\frac{m}{\epsilon}\right)}{d}}$ for every $i \in [m]$. We condition on this event for the rest of the proof. Note that for every positive neuron $j$ we have that:

$$
\begin{aligned}
\sigma(\mathbf{w}_j^\top \mathbf{x} + b_j) - \sigma(b_j) &= \sigma\left(\sum_{i \in [m]} y_i \lambda_i \sigma'_{i,j} \mathbf{x}_i^\top \mathbf{x} + b_j\right) - \sigma(b_j) \\
&\geq \sigma\left(-\lambda_s m \sqrt{\frac{2\log\left(\frac{m}{\epsilon}\right)}{d}} + b_j\right) - \sigma(b_j) \\
&\geq -\lambda_s m \sqrt{\frac{2\log\left(\frac{m}{\epsilon}\right)}{d}} .
\end{aligned}
$$

Similarly, for every negative neuron $j$ we have:

$$
\sigma(b_j) - \sigma(\mathbf{w}_j^\top \mathbf{x} + b_j) \geq -\lambda_s m \sqrt{\frac{2\log\left(\frac{m}{\epsilon}\right)}{d}} .
$$

In both bounds above we used that $\lambda_s$ is the largest among the $\lambda_i$'s. Combining both bounds, we have that:

$$
\begin{aligned}
N_{\boldsymbol{\theta}}(\mathbf{x}) &= \sum_{j=1}^{n/2} \sigma(\mathbf{w}_j^\top \mathbf{x}_r + b_j) - \sum_{j=n/2+1}^{n} \sigma(\mathbf{w}_j^\top \mathbf{x}_r + b_j) \\
&= \sum_{j=1}^{n/2} \sigma(\mathbf{w}_j^\top \mathbf{x}_r + b_j) - \sum_{j=n/2+1}^{n} \sigma(\mathbf{w}_j^\top \mathbf{x}_r + b_j) + \\
&\quad + \sum_{j=1}^{n/2} \sigma(b_j) - \sum_{j=1}^{n/2} \sigma(b_j) + \sum_{j=n/2+1}^{n} \sigma(b_j) - \sum_{j=n/2+1}^{n} \sigma(b_j) \\
&\geq \sum_{j=1}^{n/2} b_j - \sum_{j=n/2+1}^{n} b_j - \lambda_s m n \sqrt{\frac{2 \log\left(\frac{m}{\epsilon}\right)}{d}} \\
&\geq \frac{1}{4} - 3 m^2 n^2 \sqrt{\frac{2 \log\left(\frac{m}{\epsilon}\right)}{d}} .
\end{aligned}
$$

We use that $\frac{3 m^2 n^2 \log(d)\sqrt{2}}{\sqrt{d}} \leq \frac{1}{8}$ for a large enough constant $c_3$, hence $N_{\boldsymbol{\theta}}(\mathbf{x}) \geq \frac{1}{4} - \frac{1}{8} > 0$, this finishes the proof. $\qquad\square$

## C  Proofs from Section 4.2

### C.1  Proof of Proposition 4.4.

We first note that there exists $j \in \{1, 2\}$ with $v_j < 0$, since otherwise the network wouldn't be able to classify samples with a negative label (which exist by assumption). Assuming without loss of generality that $v_2 < 0$, for any $\mathbf{w}_1$ we also have that $\Pr_{\mathbf{x} \sim \mathrm{Unif}(\mathbb{S}^{d-1})}[\mathbf{w}_1^\top \mathbf{x} \leq 0] = \frac{1}{2}$. If this even occurs, then $N_{\boldsymbol{\theta}}(\mathbf{x}) \leq v_2 \sigma(\mathbf{w}_2 \top \mathbf{x}) \leq 0$.

### C.2  Proof of Proposition 4.5

Suppose $[n] = J_+ \cup J_-$ are such that

$$
N_{\boldsymbol{\theta}}(\mathbf{x}) = \sum_{j \in J_+} v_j \sigma(\mathbf{w}_j^\top \mathbf{x}) + \sum_{j \in J_-} v_j \sigma(\mathbf{w}_j^\top \mathbf{x}) ,
$$

where $v_j \geq 0$ for $j \in J_+$ and $v_j < 0$ for $j \in J_-$. Then, for any choice of $(\mathbf{w}_j)_{j \in J_+}$ :

$$
\Pr_{\mathbf{x} \sim \mathrm{Unif}(\mathbb{S}^{d-1})}[N_{\boldsymbol{\theta}}(\mathbf{x}) \leq 0] \geq \Pr_{\mathbf{x} \sim \mathbb{S}^{d-1}}[\forall j \in J_+, \ \mathbf{w}_j^\top \mathbf{x} < 0] \geq \frac{1}{2^{|J_+|}} \geq \frac{1}{2^n} .
$$

### C.3  Proof of Proposition 4.6

Assume without loss of generality that $I_- = [k]$. Consider the weights $\mathbf{w}_i = \mathbf{x}_i$, $v_i = -1$ for every $i \in I_-$, and $\mathbf{w}_{k+1} = \sum_{i \in I_+} \mathbf{w}_i$, $v_{k+1} = 1$. For this network we have $y_i N_{\boldsymbol{\theta}}(\mathbf{x}_i) = 1$, thus all points lie on the margin. In addition, it is not difficult to see that this network satisfies the other KKT conditions with $\lambda_i = 1$ for every $i \in [m]$. Note that $\Pr_{\mathbf{x} \sim \mathrm{Unif}(\mathbb{S}^{d-1})}[\mathbf{w}_{k+1}^\top \mathbf{x} < 0] = \frac{1}{2}$ and $\Pr_{\mathbf{x} \sim \mathrm{Unif}(\mathbb{S}^{d-1})}[\forall i \in I_-, \ \mathbf{w}_i^\top \mathbf{x} < 0] = \frac{1}{2^k}$, since all the $\mathbf{x}_i$ are orthogonal. Thus, we get that

$$
\Pr_{\mathbf{x} \sim \mathrm{Unif}(\mathbb{S}^{d-1})}[N_{\boldsymbol{\theta}}(\mathbf{x}) \leq 0] \geq \Pr_{\mathbf{x} \sim \mathrm{Unif}(\mathbb{S}^{d-1})}[\mathbf{w}_{k+1}^\top \mathbf{x} < 0] - \Pr_{\mathbf{x} \sim \mathrm{Unif}(\mathbb{S}^{d-1})}[\forall i \in I_-, \ \mathbf{w}_i^\top \mathbf{x} < 0] \geq \frac{1}{2} - \frac{1}{2^k} .
$$

# D  Additional probabilistic lemmas

## D.1  Proof of Lemma A.1

Clearly, the third (no collisions) condition holds almost surely, so it suffices to analyze the gaps between samples. The distribution of distances between uniformly random points on a segment is well studied [Pyke, 1965, Holst, 1980] and the lemma can be derived by known results. Nonetheless we provide a proof for completeness.

Denote by $\Delta_1 \leq \Delta_2 \leq \cdots \leq \Delta_{m+1}$ the *ordered* spacings $x_1, (x_2-x_1), \ldots, (x_m-x_{m-1}), (1-x_m)$. With this notation, note that $E_x$ occurs if $\Delta_{m+1} \leq \frac{\log(8(m+1)/\delta)}{m+1}$ and $\Delta_{m/8} \geq \frac{1}{10m}$. We will show that each of these conditions holds with probability at least $1 - \frac{\delta}{4}$, under which we would conclude by the union bound. Let $Z_1, \ldots, Z_{m+1} \overset{iid}{\sim} \mathrm{Exp}(1)$ be unit mean exponential random variables, and denote their ordering by $Z_{(1)} \leq \cdots \leq Z_{(m+1)}$. The main well known observation which we use is that for any $j \in [m+1]: \ \Delta_j \overset{d}{=} \frac{Z_{(j)}}{\sum_{i=1}^{m+1} Z_i}$ (see Holst, 1980). Hence for any $t > 0$:

$$\Pr[(m+1)\Delta_j - \log(m+1) \leq t]$$

$$= \Pr\left[ Z_{(j)} - \log(m+1) \leq t + (t + \log(m+1))\left( \frac{\sum_{i=1}^{m+1} Z_i}{m+1} - 1 \right) \right].$$

Note that $\frac{\sum_{i=1}^{m+1} Z_i}{m+1} - 1 \overset{p}{\longrightarrow} 0$ as $\mathbb{E}\left[ \frac{\sum_{i=1}^{m+1} Z_i}{m+1} \right] = 1$ and $\mathrm{Var}\left[ \frac{\sum_{i=1}^{m+1} Z_i}{m+1} \right] = \frac{m+1}{(m+1)^2} \overset{m\to\infty}{\longrightarrow} 0$.

Furthermore,

$$\Pr\left[ Z_{(j)} - \log(m+1) \leq t \right] = \Pr\left[ Z_{(j)} \leq t + \log(m+1) \right] = \Pr[Z_{(1)}, \ldots, Z_{(j)} \leq t + \log(m+1)]$$

$$= \left( 1 - e^{-t-\log(m+1)} \right)^j = \left( 1 - \frac{e^{-t}}{m+1} \right)^j.$$

By introducing the change of variables $r = \frac{t+\log(m+1)}{m+1}$ we conclude that

$$\lim_{m\to\infty} \Pr[\Delta_j \leq r] = \left( 1 - \frac{e^{-(m+1)r+\log(m+1)}}{m+1} \right)^j.$$

It remains to plug in our parameters of interest. For $j = m+1$ we get

$$\lim_{m\to\infty} \Pr[\Delta_{m+1} \leq r] = \lim_{m\to\infty} \left( 1 - \frac{e^{-(m+1)r+\log(m+1)}}{m+1} \right)^{m+1} = \lim_{m\to\infty} \exp\left( -e^{-(m+1)r+\log(m+1)} \right).$$

Noting that for $r = \frac{8\log(8(m+1)/\delta)}{m+1}$ it holds that $e^{-(m+1)r+\log(m+1)} \to 0$ so we can use the Taylor approximation $\exp(z) \approx 1 + z$ and conclude that

$$\lim_{m\to\infty} \Pr[\Delta_{m+1} \leq r] = 1 - e^{-(m+1)r+\log(m+1)} = 1 - \frac{\delta}{8},$$

where the last equality holds for $r = \frac{\log(8(m+1)/\delta)}{m+1}$. Overall for $m$ larger than some numerical constant, the left hand side is larger than $1 - \delta/4$ as required.

The second condition follows a similar computation for $j = \frac{m+1}{8}$, $r = \frac{\log(m/8\log(8/\delta))}{8(m+1)}$, while using the fact $\Pr\left[ \Delta_{m/8} > \frac{1}{10(m+1)} \right] = 1 - \Pr\left[ \Delta_{m/8} \leq \frac{1}{10(m+1)} \right]$.

## D.2  Concentration bounds for vectors on the unit sphere

Here we present some standard concentration bounds for uniformly sampled points on the unit sphere in high dimension.

**Lemma D.1** (Lemma 2.2 from Ball, 1997). $\Pr_{\mathbf{u},\mathbf{v}\sim\mathrm{Unif}(\mathbb{S}^{d-1})}[|\mathbf{u}^\top \mathbf{v}| \geq t] \leq 2\exp\left( \frac{-dt^2}{2} \right)$.

**Lemma D.2** (Exercise 4.7.3 from Vershynin, 2018). *Let $\mathbf{x}_1, \ldots, \mathbf{x}_m \sim \mathrm{Unif}(\mathbb{S}^{d-1})$, and denote by $X$ the matrix whose $i$'th row is $\mathbf{x}_i$. Then, there is a universal constant $c > 0$ such that:*

$$\Pr\left[ \|XX^\top - I\| \geq \frac{c}{d} \cdot \left( \sqrt{\frac{d+t}{m}} + \frac{d+t}{m} \right) \right] \leq 2e^{-t}.$$

