# OpenReview forum: "From Tempered to Benign Overfitting in ReLU Neural Networks"
_NeurIPS.cc/2023/Conference — NeurIPS 2023 spotlight_

### Official Review · Reviewer_s2UC · 2023-07-01

**Soundness:** 3 good
**Presentation:** 4 excellent
**Contribution:** 3 good
**Rating:** 6
**Confidence:** 4

**Summary:**

This work examines asymptotic overfitting behavior on a simple class of toy classification problem for two-hidden-layer ReLU nets, finding that increasing input dimension tends to push the model from tempered towards benign overfitting.

**Strengths:**

Originality: this is, to my knowledge, the first analysis of tempered overfitting for nets outside the kernel regime. The broader problem's admittedly decently well-studied, and the conclusions here are more or less in line with intuitions from other works, but the authors are doing something new here.

Quality: while I haven't checked the proofs, the theoretical results seem conceptually solid. While they rely on certain assumptions (e.g., assuming you've reached a KKT point or local minimum), the authors are upfront about these. I have some Qs about the experiments below.

Clarity: the paper's very well organized and written. Most things were clear on a single linear readthrough.

Significance: to my mind, the nice takeaways here are (a) the identification of a very simple toy problem for the study of these regimes and (b) theoretical results for classification that complement those of Mallinar et al. (which only study (kernel) *regression*). It's nice that the authors recover the $\Theta(p)$ scaling in their Theorem 3.2. I hadn't seen the cited results about margin maximization used in a setting like this; it's nice that they let one avoid consideration of the network dynamics entirely.

**Weaknesses:**

One omission here seems to be comparison to the experiment of Figure 9 of Mallinar et al, which shows that Laplace kernels basically exhibit the core phenomenon found by the paper: i.e., larger input dim pushing fitting from tempered to benign (with no need to scale dim w.r.t. dataset size).

I suppose the biggest weakness here is that this paper's results seem like exactly what you'd intuitively expect from (the much simpler model of) kernel regression, and so it's sort of unclear what new thing we've gained from this long set of calculations. (For the purposes of review, I wouldn't count this *against* the work -- and I do appreciate that establishing facts for neural nets is hard, and improving techniques there is useful.) It's also sort of unclear to me that the proof techniques used here will scale to other problems -- they seem clever but ad-hoc, and the tempered results rely crucially on the input space being 1D -- and so it's not obvious that, e.g., the specialized + expected results here will enable extension to other more general + powerful conclusions later.

The empirics seem generally good, but a bit odd in certain ways. For example, the input dimensions are very large! The aforementioned Figure 9 of Mallinar et al. suggests that much smaller input dims should be needed to see the correct behavior here. (Could be due to the difference between classifcation + regression; see note A below). A bigger oddity's that the difference in training set size between the two plots of Figure 1 is only 4x, which is quite small. It seems like the authors are trying to say that maybe "when [input dim] >> [dataset size], fitting is benign, and vice versa," but if that's the message, there are much more compelling ways of showing that, including having much greater variation in both parameters or making a 2D heatmap at some fixed and intermediate $p$.

Another concern with the empirics is that converging can presumably take a very long time when you're relying on the exponential tail of the loss fn to push you. Seems worth confirming that these experiments won't change after a much longer training time.

**Questions:**

Could the authors explain the definition of "local minimum" before Theorem 3.2? I'm confused as to how points satisfying this condition are local minima.

Note A: as an aside, just to flag it, a potentially interesting Q here seems like studying the difference between classification and regression. I suspect that Figure 1 would look different in these two cases. (In particular, for regression, the results of Mallinar et al. for KR suggest that you'd see a linear fit whose *slope* depends on input dim, whereas for classification you see this quite complicated nonlinear shape that intersections with the line $y=x$ at the endpoints. I wonder what the shape of that curve is.) This Q is AFAIK totally unstudied, even for KR, for which is seems probably tractable.

---

> ### Author Rebuttal · Authors · 2023-08-07
>
> We thank the reviewer for the thorough review and constructive comments.
>
> - "Comparison to the experiment of Figure 9 of Mallinar et al.": This is indeed an interesting comparison, although we point out that we analyze a classification setting, while the aforementioned experiment deals with regression. We agree it would be very interesting to examine the relations between these two settings, though this is beyond the scope of our paper.
>
> - "This paper's results seem like exactly what you'd intuitively expect from... kernel regression": As far as we know, our work gives the first theoretical result for tempered overfitting in neural networks - which even if "confirming prior beliefs", is interesting by its own right in our opinion. Moreover,  we note that our work is on classification, while most previous works (for kernels) examined regression settings.
>
> - "The empirics seem generally good, but a bit odd in certain ways": We agree that it is possible to do a more thorough empirical study, as was done by Mallinar et al., although this is not the main focus of our paper.
>
>
> - " ...converging can presumably take a very long time when you're relying on the exponential tail... ": In our experiments we trained for $20k$ epochs (line 375), which is much longer than it takes to reach an accuracy of $100$%.  We also tested for longer training times ($100k$ epochs) and the results were not affected.
>
> - "Definition of local minimum before Theorem 3.2": This is the standard definition of a local minimum under constraints, as opposed to a local minimum in the general unconstrained sense. In other words, $\theta$ is a local minimum of a function $f$ under some constraints, if there exists a neighborhood containing it, such that every $\theta'$ in this neighborhood which also satisfies the constraints satisfies $f(\theta')\geq f(\theta)$. Note that in our case, $f$ is just the $L_2$ norm function, while the constraints are interpolating the dataset. We will make further attempts to clarify this issue in the final version.

---

### Official Review · Reviewer_dEM1 · 2023-07-07

**Soundness:** 4 excellent
**Presentation:** 4 excellent
**Contribution:** 4 excellent
**Rating:** 7
**Confidence:** 4

**Summary:**

This paper considers the various settings of overfitting, including benign overfitting where an interpolating estimator is optimal in spite of perfectly fitting noisy labels, tempered overfitting where an interpolating estimator is inconsistent but error is a function of the label noise, and catastrophic overfitting where an interpolating estimator has no ability to generalize (random guessing in classification or infinite risk in regression). The authors follow-up prior works in benign overfitting by showing that a two layer neural network can benignly overfit so long as the input dimension of the data grows faster than the number of training samples, irrespective of the width of the network itself (assuming convergence to a max-margin solution, and additionally show another benign overfitting result that only requires convergence to a KKT point of a max-margin problem).

They also provide a theoretical justification for tempered overfitting in two-layer ReLU networks, a phenomena that was empirically studied in prior work. The authors precisely characterize the excess risk of an interpolating network trained to fit uniform data with labels samples as +1 and flipped to -1 with probability p. The authors show that for one-dimensional data, the excess risk of such a network interpolating this problem is tightly bounded by functions of the noise level (e.g. theta(poly(p)) and theta(p) in a more restrictive setting).

The authors then observe that catastrophic overfitting can occur when forcing networks, of width at least 2, without a bias to interpolate the training data (drawn as specified before) when there is at least one sample with -1 label. They also show that you cannot obtain benign overfitting from convergence to a KKT point of the max-margin problem unless the network has a bias term.

Finally, the authors provide an extended version of an experiment from prior work on tempered overfitting in which fully-connected ReLU networks are trained to interpolate data on the unit sphere with +1/-1 labels sampled with probability p, as defined before. They observe that there are regions between benign and tempered that are a function of the ratio of input data dimension to number of training samples, namely: as the input data dimension grows higher than the number of training samples, the excess risk of the model approaches 0. They additionally show experimental results on 3-layer networks to empirically assert that, while their theory holds for 2 layer networks practically the same behavior is seen for 3. Finally, they experimentally show in their setting that the width of the network is not as important as the input dimension when evaluating consistency of the estimator.

EDIT: Thank you for the rebuttal and engaging response, this is indeed a very thoughtful work. I opt to keep my positive score.

**Strengths:**

This paper provides a clear theoretical analysis for tempered overfitting for two-layer ReLU networks that has not appeared in prior literature. There is some earlier theoretical work on tempered overfitting in kernel regression and for minimum description length settings, but as far as I know nothing for a standard finite-width neural network. The prior cited work (Mallinar et al. 2022) shows experimental evidence with neural networks that suggests tempered overfitting, and following their formalisms the authors of this work go a step further to give proofs of excess risk of one of the experiments in that paper (more on this in later section).

I really appreciate the authors providing proof outlines and intuitions in the main text, as well as citing references where they are using techniques and following proof styles introduced in prior works. The style of explanation is accessible and gives me, at a high level, the right amount of information to contextualize the theorem statement while deferring details to the appendix should I want to review them.

There are many intriguing results from this paper. For one, I am initially surprised by the one-dimensional tempered results and Theorem 4.1 on benign overfitting feature bounds that don’t depend on the width of the network. I’ve written more about this in the questions section, but I would have expected a more subtle interplay between width of the network and input dimension of the data. Notably, though, the width of the network has to be sufficiently large to interpolate the data (e.g. overparameterized) and since the authors study 2-layer networks this might imply a large width, though their experimental results show roughly similar behavior for width 3000 and width 50 networks) across varying input dimensionality.

I find the role of bias section illuminating as well. To the best of my knowledge, I haven’t seen a generic negative result like this, that any bias-less network will have a risk lower bounded by a function of the network width or the support of the negative training samples.

**Weaknesses:**

The authors do acknowledge it and give a tighter bound subsequently under stronger assumptions, but Theorem 3.1 has an upper and lower bound gap that is growing with p. It’s not really a weakness of the paper, but it would be nice if the authors wrote a bit about when this bound can be very tight (as a function of p, m?).

Theorem 4.3 seems to indicate a dependence on p being smaller than some constant divided by the width of the network squared, which seems to indicate for very wide networks that this result holds for p shrinking -> 0. While interesting, I’m not sure if this is the most general result and the authors don’t discuss in too much detail the implications of this, and what settings of network width lead to various results apart from n=O(1). For instance, it could be useful to assert if p is meaningful for n >> m or n ~ m, etc. There are common settings of network width with relation to number of samples or input dimensionality and at first glance I don’t know whether this theorem will give me a useful result in many of those settings.

Additionally, this theorem trades off one assumption (convergence to a max-margin solution vs. KKT point of max-margin problem) for another (output weights are fixed as +- 1 and only the input weights are trained). They mention this theorem stems from attempting to drop the strong assumptions of the prior theorem (Theorem 4.1), but to me it seems perhaps to still include a new, strong assumption. Granted, I’m unfamiliar with the works cited in the footnote that study this setting in particular. Maybe the authors can provide a little exposition about why this setting is of particular interest and what motivates it in the grander study of neural networks?

Missing concluding thoughts, would like to see some final discussion on everything and where the authors see next steps going.

Other notes / fixes:

- Page 5, 3rd paragraph: double “we we” in the line “we we get that the total length of all…”
- Figures could use grid-lines as well as separate line-styles to differentiate the settings for accommodation to colorblind people and for printing in black and white

**Questions:**

In the setting where all layers are trained the width does not seem to be so important. This definitely speaks to the intuition I have that the ambient dimension of the data is what matters. When studying a random feature model the ambient dimension is indicated by the width of the first layer, but when you train that layer too the ambient dimension would be the input dimension. These results make me want to understand more the interplay between intrinsic and ambient dimension, as well as bottleneck vs. non-bottleneck networks in overfitting. Do you have any further intuitions about these comparisons? Is this width-independence something you would expect to hold for deeper networks, or other architectures?

Does Proposition 4.5 subsume Proposition 4.4? Could you explain a bit more about the difference here.  n=2 in Prop 4.5 would lead to a lower bound of ¼ but in Prop 4.4 the lower bound is ½. I haven’t read deeply the proofs on this section but it might help to give a little more exposition on what assumptions are made about width that differ between 4.4 and 4.5.

You prove tempered overfitting for one-dimensional data only, but practically it seems that it is more about certain settings where the ratio of input dimension to number of samples satisfies some criteria (at least, experimentally and based on prior works). Can you speak to the difference between the theorem you propose and practice? It appears that tempered overfitting could occur at any input dimension, depending on how much training data is used. Can your proof techniques be extended to explain this?

---

> ### Author Rebuttal · Authors · 2023-08-07
>
> We thank the reviewer for the very thorough review, constructive comments and positive feedback.
>
> - Bound for Theorem 3.1: We acknowledge that this bound is not tight (even for very large values of $m$), while the bound in Theorem 3.2 is tight yet requires stronger assumptions. We believe that the non-tightness of the bound is an artifact of the proof technique, and may be improved using a different analysis.
>
> - Theorem 4.3 and dependence of $p$ on $n$: We believe that this is also an artifact of the proof technique, since we only used the relatively-weak assumption of convergence to a KKT point, instead of convergence to a local or global minimum of the max-margin problem. Removing this dependence may require a more intricate analysis, or an assumption on "which" KKT point the network converge to. We note that in our experiments there is indeed no dependence of $p$ on $n$, which indicates that this assumption might be removed via a different analysis.
>
> - Assumption on fixed output weights: This setting is mostly used to make the analysis more tractable, hence it is used in many different works even beyond the topic of benign overfitting (e.g. for analyzing optimization in neural networks). We will elaborate the exposition on this assumption, and include some citations of related works.
>
> - Concluding thoughts: Thank you for these suggestions. We will add a conclusion section to the camera ready version if space permits, and make the figures more accessible to colorblind people by adding grid-lines and different line styles.
>
> - "Is this width-independence something you would expect to hold for deeper networks, or other architectures?":
> We do expect that width independence will hold for other architectures, under the assumption of convergence to a solution of the max-margin problem. In fact, such convergence indicates that the norm of all the weights is bounded, and adding more neurons shouldn't affect this bound. This may imply that the norm of the weights has a larger effect on the analysis, than the width, or number of neurons -- as indicated by many works on the sample complexity of NNs. We will elaborate on this issue in the final version.
>
> - "Does Proposition 4.5 subsume Proposition 4.4?" Although they are similar, Proposition 4.5 does not subsume Proposition 4.4. Note that the bound from Proposition 4.5 would only show a lower bound of $\frac{1}{4}$, while the lower bound in Proposition 4.4 is $\frac{1}{2}$, which falls under the definition of catastrophic overfitting. We will clarify and emphasize this issue in the revision.
>
> - "You prove tempered overfitting for one-dimensional data only...": We do believe there is a more intricate connection between the input dimension and number of training samples for intermediate dimensions. We also think that our proof techniques (analyzing solutions or KKT points of the max-margin problem) can be extended to higher dimensions, although the analysis is much more intricate. We can currently prove these results only for 1-dimensional or very high dimensional data, and extending our techniques to intermediate dimensions is a very interesting (and possibly challenging) future direction.

---

### Official Review · Reviewer_2xri · 2023-07-07

**Soundness:** 4 excellent
**Presentation:** 4 excellent
**Contribution:** 3 good
**Rating:** 6
**Confidence:** 4

**Summary:**

This work studies the phenomenon of overparameterized neural networks (NNs) generalizing well even when trained on noisy data. Previous research focused on "benign overfitting," where interpolating predictors achieve near-optimal performance. However, recent empirical observations suggest that NN behavior is better described as "tempered overfitting," with non-optimal yet non-trivial performance that degrades with increasing noise levels. In this study, the authors provide theoretical justification and empirical validation, showing that the type of overfitting transitions from tempered to benign as the dimensionality increases in a simple classification setting with 2-layer ReLU NNs. Their findings highlight the intricate connections between input dimension, sample size, architecture, training algorithm, and resulting overfitting.

**Strengths:**

This paper establishes the connection between data dimensionality and the occurrence of benign or tempered overfitting. It presents three theorems that address benign, tempered, and catastrophic overfitting, respectively, corresponding to one-dimensional data, high-dimensional data, and intermediate-dimensional data.

The study focuses on a two-layer ReLU neural network that includes a bias term, which is a more comprehensive approach compared to recent works that examined the two-layer ReLU neural network without a bias term. Additionally, a specific data distribution is considered in this paper. Furthermore, the paper explores the relationship between the bias term and the occurrence of benign/tempered overfitting.

**Weaknesses:**

The findings presented in this paper are built upon the outcomes (specifically, the convergence to the KKT point of the max-margin problem) obtained in previous studies by Lyu and Li (2020) and Ji and Telgarsky (2020). To be more precise, the results in this paper depend on either achieving convergence to a KKT point or making stronger assumptions, such as converging to a local optimum. Nonetheless, these assumptions are reasonable considering that numerous other papers also derive their results based on the convergence to the KKT point, such as the work by Frei et al. (2023).

It is important to note that this paper exclusively focuses on a simplistic data model, where the samples originate from a unit sphere and the labels are fixed constants, specifically +1. The discussion does not extend to other more prevalent data distributions.

**Questions:**

It would be particularly intriguing to investigate whether these findings remain consistent when applied to other commonly studied data distributions, such as mixture distributions. Additionally, it would be valuable to explore the possibility of extending the analysis techniques utilized in the proof to encompass a broader range of distributions.

**Limitations:**

Same as weaknesses.

---

> ### Author Rebuttal · Authors · 2023-08-07
>
> We thank the reviewer for the constructive review and insightful comments.
>
> We used this data distribution as it was already studied in previous works (e.g. Mallinar et al. 2022), while making the analysis tractable. Our results for one-dimensional data can be readily extended for other distributions using the same proof technique. On the other hand, it seems that the results for high-dimensional data may require a more intricate analysis for such extensions, which we leave for future research. We agree that it would be interesting to further use our techniques to study a broader range of distributions. Moreover, we also believe that our techniques can be used to further study other phenomena, beyond benign and tempered overfitting, by utilizing the implications of the implicit bias of gradient methods towards margin maximization.

---

### Official Review · Reviewer_M9si · 2023-07-15

**Soundness:** 4 excellent
**Presentation:** 4 excellent
**Contribution:** 3 good
**Rating:** 8
**Confidence:** 3

**Summary:**

This paper explores the spectrum of benign to catastrophic overfitting in a specific instance of a 2-layered neural network with ReLU activation when the training data has noisy labels with proportion $p$. The authors when the input dimension is , the test error is lower bounded proportionally to $poly(p)$ under different assumptions. When the input dimension is high, i.e. $d=poly(n)$, a NN overfitting (i.e. achieving perfect training accuracy) is benign and good generalization performance is exhibited. The authors also consider the intermediate  regime for the input dimension and show a gradual behavior for the error rate as a function of the noise factor.


**Strengths:**

1. The paper is very well written with a great exposition and tradeoff between results and intuition.
2. Results presented seem comprehensive and I like the fast the authors do not neglect the intermediate regime and provide empirical evidence where theoretical analysis is not attainable.
3. The authors make their theoretical results accessible and explain the importance of the bias for the analysis.

**Weaknesses:**

1. The paper is missing a discussion/conclusion section to tie the results to real scenarios. Consider shortening one of the sections/proof sketches in favor or the above.
2. The assumptions of the theoretical results are not very realistic, specifically in theorem 4.3, the input dimension is big with respect to both m and n, this does not follow common practice of deep learning where the number of parameters exceeds the number of examples but usually the input dimension is not very big.
3. The considered architecture is rather limited, it would be interesting to explore empirically with much greater depth and not only 2 and 3.

**Questions:**

1. Is the assumption that the ground truth labels are 1 and all the noise is labeled -1 necessary for the analysis or just a simplifying assumption for presentation?
2. The authors mention that the width of the network probably doesn't play an important role in the benign overfitting phenomena - is this also the case for more realistic empirical setups? it seems to contradict common practice of deep learning where the number of parameters scales and benign overfitting is exhibited.

**Limitations:**

The assumptions made in the theoretical setup are not very realistic, that being said, the reviewer appreciates the difficulty in obtaining such theoretical results. I would recommend that the authors discuss the assumptions and limitations in a dedicated section.

---

> ### Author Rebuttal · Authors · 2023-08-07
>
> We thank the reviewer for the thorough review, helpful comments and positive feedback.
>
> Weaknesses:
>
> - 1. If space permits, we will consider adding a short discussion on practical implications of our results. We note that such discussions also appear in several previous papers on benign and tempered overfitting (e.g. Bartlett et al. 2019, Mallinar et al. 2022).
> - 2. We emphasize that in most papers regarding benign overfitting the input dimension is much larger than the number of training samples, e.g. Bartlett et al. 2019, Shamir 2022. This is arguably a valid drawback shared by this whole line of work.
> - 3. We indeed agree that it would be interesting to study empirically more architectures and different settings under which benign/tempered overfitting holds. However, in this paper we focused more on the theoretical aspects, and we leave a thorough empirical investigation for future work.
>
> Questions:
> - 1. This is an interesting question. We used this data distribution as it was already studied in previous works (e.g. Mallinar et al. 2022), while making the analysis tractable. Our results for one-dimensional data can be readily extended for other distributions using the same proof technique. On the other hand, it seems that the results for high-dimensional data may require a more intricate analysis for such extensions, which we leave for future research.
>
> - 2. We believe that also in practical settings, the width doesn't play a role in the type of overfitting that appears, as long as the network is large enough to interpolate the data. Note that for our high-dimensional setting, the number of trainable parameters increases with the input dimension, which we assume is larger than the number of samples. Hence, this ensures the network can interpolate the data, with no restrictions on the width. Notably, a parallel line of work establishes advantages in terms of *optimization* for wider networks (e.g. NTK analyses), which is not the focus of this paper in which perfect interpolation is assumed throughout.

---

> > ### Comment · Reviewer_M9si · 2023-08-19
> > **Response to rebuttal**
> >
> > The reviewer appreciates the detailed response by the authors, all concerns and questions have been addressed.
> >
> > The reviewer maintains the positive stance towards the paper - this is a strong submission worthy of publication.

---

### Decision · Program_Chairs · 2023-09-21

**Decision:**

Accept (spotlight)

**Comment:**

This paper theoretically studies the phenomenon of "benign overfitting", in a certain simplified setting. They find that for two-layer ReLU networks, increasing the input dimension can shift the model from the tempered to the benign regime.
Reviewers were enthusiastic about the strength of the work, and the clarity of the writing. Extending the theoretical study of benign overfitting beyond kernels is a valuable contribution. Thus, I recommend acceptance.